# Lateral hypothalamic glutamatergic inputs to VTA glutamatergic neurons mediate prioritization of innate defensive behavior over feeding

M. Flavia Barbano [1], Shiliang Zhang [2], Emma Chen[1,3], Orlando Espinoza [1], Uzma Mohammad[1], Yocasta Alvarez-Bagnarol[1,4], Bing Liu[1], Suyun Hahn[1] & Marisela Morales [1] ✉

The lateral hypothalamus (LH) is involved in feeding behavior and defense responses by interacting with different brain structures, including the Ventral Tegmental Area (VTA). Emerging evidence indicates that LH-glutamatergic neurons infrequently synapse on VTA-dopamine neurons but preferentially establish multiple synapses on VTA-glutamatergic neurons. Here, we demonstrated that LH-glutamatergic inputs to VTA promoted active avoidance, long-term aversion, and escape attempts. By testing feeding in the presence of a predator, we observed that ongoing feeding was decreased, and that this predator-induced decrease in feeding was abolished by photoinhibition of the LH-glutamatergic inputs to VTA. By VTA specific neuronal ablation, we established that predator-induced decreases in feeding were mediated by VTA-glutamatergic neurons but not by dopamine or GABA neurons. Thus, we provided evidence for an unanticipated neuronal circuitry between LH-glutamatergic inputs to VTA-glutamatergic neurons that plays a role in prioritizing escape, and in the switch from feeding to escape in mice.

Reproduction and survival at the individual and species levels depend on a wide range of goal-directed and innate behaviors, such as feeding, drinking, mating, and defensive responses. Proper action selection between competing, usually mutually exclusive, innate behaviors is critical for survival fitness and relies on neural integration of external stimuli and internal states[1,2]. Feeding and defense are critical innate behaviors across organisms, of which selection depends on both environmental factors (such as resource availability, density of predators, or shelter accessibility) and internal states (such as hunger or illness). Indeed, early studies have demonstrated that when animals are confronted with threatening

stimuli or environments, they decrease eating behavior (hypophagia), which is accompanied by a reduction in the release of digestive secretions[3]. Fear-induced hypophagia may be elicited by innate stimuli, such as the odor of a predator[4,5], or by learned stimuli, such as a tone previously associated with foot-shock delivery[6,7]. In contrast, when animals are food deprived, they are more likely to engage in riskier behaviors such as enhanced exploration and farther foraging from a shelter[8,9]. A brain structure strategically located to sense internal states and to coordinate physiological survival responses is the hypothalamus[10], and specifically its subnucleus, the lateral hypothalamus (LH). The LH plays a critical role in integrating

[1]Integrative Neuroscience Research Branch, National Institute on Drug Abuse, National Institutes of Health, Baltimore, MD 21224, USA. [2]Confocal and Electron Microscopy Core, National Institute on Drug Abuse, National Institutes of Health, Baltimore, MD 21224, USA. [3]Present address: Rutgers New Jersey Medical School, Newark, NJ 07103, USA. [4]Present address: Department of Anatomy and Neurobiology, University of Puerto Rico, Medical Sciences Campus, San Juan, Puerto Rico, USA. ✉e-mail: MMORALES@intra.nida.nih.gov

endocrine, autonomic, and behavioral responses related to basic behaviors, such as feeding, drinking, sex, and defense[11–14].

The LH is composed of genetically and functionally distinct neural subpopulations, including glutamatergic and GABAergic neurons[13,15,16], targeting different brain structures[17–22], including the ventral tegmental area (VTA), which receives inputs from both LH-GABAergic and LH-glutamatergic neurons[19–22]. Given that the VTA contains dopaminergic, GABAergic, and glutamatergic neurons[23], recent studies have addressed the role that circuitry involving LH-GABAergic and LH-glutamatergic neurons synapsing on different subpopulations of VTA neurons plays in behavior[22,24–27].

LH-GABAergic inputs to the VTA have been proposed to play a role in feeding and reward[24–26]. Based on electrophysiological recordings and measurements of dopamine release in the nucleus accumbens, a model has been proposed in which LH-GABAergic neurons, via their synapses on VTA-GABAergic neurons, disinhibit VTA-dopamine neurons, mediating feeding and reward[25,26]. In contrast, LH-glutamatergic inputs to VTA-dopamine neurons have been proposed to play a role in place avoidance[26] and in encoding responses to stimuli predicting aversive outcomes[27]. In addition to the above-proposed circuitry between LH-GABAergic and VTA-GABAergic neurons and between LH-glutamatergic and VTA-dopaminergic neurons, we have recently demonstrated that within the VTA, LH-glutamatergic neurons preferentially establish excitatory synapses on glutamatergic neurons and that this pathway plays a role in innate defensive behaviors[22]. Collectively, these findings indicate that specific types of synapses between LH-glutamatergic or LH-GABAergic neurons and diverse postsynaptic phenotypes of VTA neurons are critical in the regulation of feeding and defensive escape.

Here, by a multidisciplinary approach, we further characterized the role of LH-glutamatergic neurons innervating the VTA in aversion, active avoidance, escape behavior, and predator-induced decreases in feeding behavior. In addition, we investigated the extent to which LH-glutamatergic inputs to the VTA play a role in action selection between innate feeding and escape behaviors. We uncovered an unpredicted glutamatergic circuit between the LH and VTA-glutamatergic neurons that play a critical role in selecting defensive escape over feeding behavior.

## Results

### VTA release of glutamate from LH-VGluT2 fibers induces aversion and escape attempts mediated by the activation of VTA glutamate receptors

We targeted LH-VGluT2 neurons and their axons by injecting a Cre-inducible adeno-associated virus (AAV) with a double-floxed inverted open reading frame (DIO) expressing channelrhodopsin-2 (ChR2; ChR2-eYFP mice) tethered to an enhanced yellow fluorescent protein (eYFP) in the LH of VGluT2::Cre mice (Fig. 1A, Supplementary Fig. 1A, B, F). Control mice were injected with a vector lacking ChR2 (eYFP mice), optic fibers were implanted dorsal to the VTA of ChR2-eYFP or eYFP mice (Fig. 1A, Supplementary Fig. 1A–C, F, G), and we detected high expression of eYFP fibers without somatic labeling in the VTA of eYFP or ChR2-eYFP mice (Supplementary Fig. 2A–K). By cFos quantification, we found that VTA optical stimulation of LH-VGluT2 fibers induced local cFos expression in ChR2-eYFP mice (Supplementary Fig. 2L, M). By testing eYFP and ChR2-eYFP mice in an operant chamber, in which rotation by a quarter turn of the active wheel resulted in VTA photostimulation (Supplementary Fig. 3A), we found that the number of reinforced quarter wheel turns did not vary as a function of pulse durations or frequencies (Supplementary Fig. 3B). In addition, while the number of reinforced quarter wheel turns at 20 Hz was similar between groups (Supplementary Fig. 3C), the total quarter

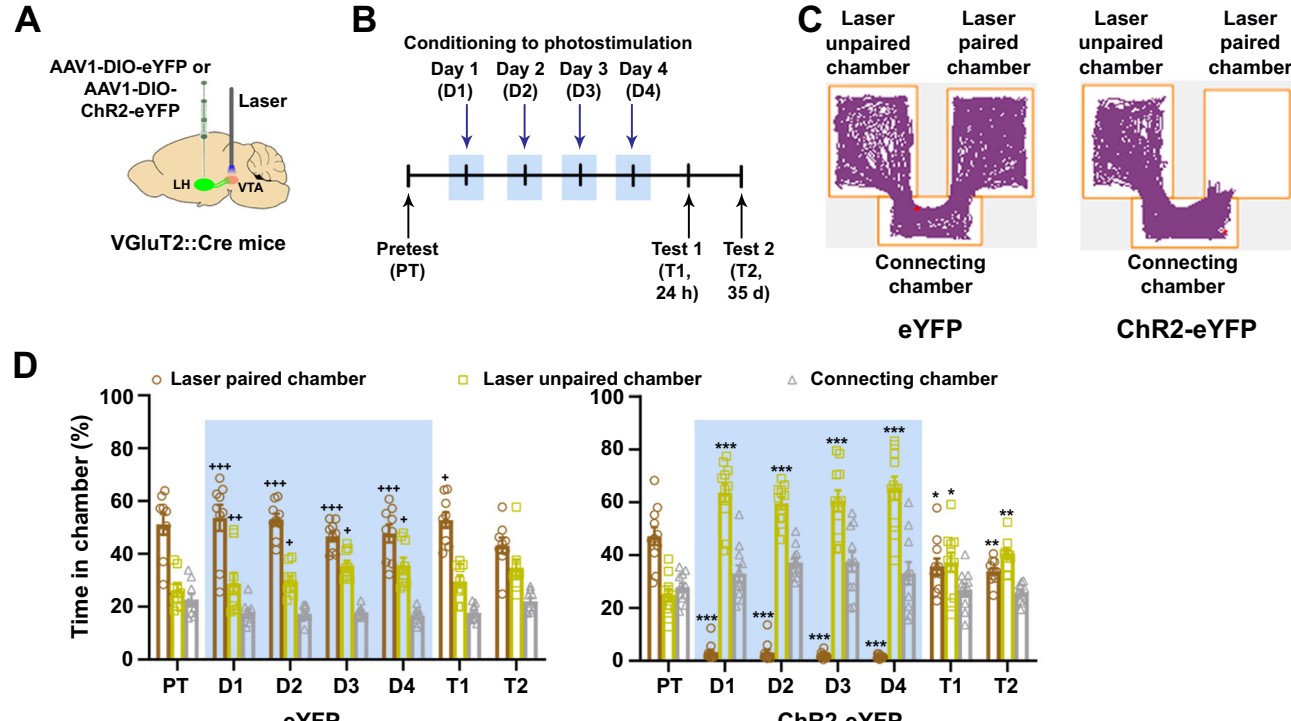

**Fig. 1 | Release of glutamate from LH-VGluT2 fibers within the VTA induces aversion. A** LH injection of AAV1-DIO-eYFP or AAV1-DIO-ChR2-eYFP and VTA optic fiber. **B** Timeline for place conditioning experiments. **C** Track plot for an eYFP and a ChR2-eYFP mouse during the first day of photostimulation conditioning (D1). **D** ChR2-eYFP mice (*n* = 11) spent significantly less time in the laser-paired chamber than eYFP control mice (*n* = 9) during and after photostimulation sessions

(group × chamber × experimental phase: $F_{(12,216)} = 21.46$, $P < 0.00001$, ANOVA with Newman-Keuls post hoc test). * $P < 0.05$, ** $P < 0.01$, *** $P < 0.001$, against pretest; + $P < 0.05$, ++ $P < 0.01$, +++ $P < 0.001$, against ChR2-eYFP. Light-blue rectangles indicate photostimulation. Data are presented as mean values ± SEM. Source data are provided as a Source Data file.

wheel turns on the active wheel remained significantly lower in ChR2-eYFP mice throughout training (Supplementary Fig. 3D). These findings indicate that VTA photostimulation of LH-VGluT2 fibers is not reinforcing, but instead, is aversive.

In a follow up study, we tested ChR2-eYFP and eYFP mice in a three-chamber apparatus in which one of the chambers was paired with photostimulation, and each time mice entered the laser-paired chamber, they received continuous trains of VTA photostimulation at 20 Hz. We observed that ChR2-eYFP mice, but not eYFP mice, displayed significant active avoidance to the laser-paired chamber from the first to the fourth photostimulation conditioning session (Fig. 1B–D). This avoidance by ChR2-eYFP mice to the laser-paired chamber was observed in the absence of VTA photostimulation 24 h and 35 days after the last conditioning session (Fig. 1D). In contrast, we found that while ChR2-eYFP mice tested at a frequency of 2.5 Hz (instead of 20 Hz) avoided the laser-paired chamber during training and 24 h after training (Supplementary Fig. 4A–D), the avoidance was absent 35 days after the last conditioning session (Supplementary Fig. 4D). These findings indicate that VTA activation of LH-VGluT2 fibers drives aversion and that the duration of this aversion depends on the conditions by which these LH-VGluT2 fibers are activated in the VTA.

We next applied a pharmacological approach to evaluate the participation of VTA glutamatergic receptors in the place avoidance elicited by VTA photostimulation of LH-VGluT2 fibers in ChR2-eYFP mice. By intra-VTA injections, we administered artificial cerebrospinal fluid (aCSF) or a mix of the NMDA receptor antagonist (AP5; 5 µg/µl) and the AMPA receptor antagonist (CNQX; 5 µg/µl), in the absence (Supplementary Fig. 5A, B, D) or presence of photostimulation 3 min prior to testing mice in the chamber apparatus (Fig. 2A, B). We found that in the absence of photostimulation, the mix of NMDA and AMPA receptor antagonists did not induce avoidance behavior (Supplementary Fig. 5B), but it blocked the avoidance induced by VTA photostimulation (Fig. 2C). These findings indicate that avoidance responses elicited by VTA photostimulation of LH-VGluT2 fibers are mediated by VTA glutamate receptors.

Given that prior studies have shown that glutamatergic projections from LH to either the lateral habenula[28,29] or the periaqueductal gray[30] induce aversion, there is the possibility that backpropagation of action potentials from stimulated LH-VGluT2 axons in the VTA might travel to the LH cell bodies, resulting in the activation of LH axon collaterals projecting outside the VTA. Thus, we determined whether induced avoidance responses after VTA photostimulation of LH-VGluT2 fibers involved LH-VGluT2 neuronal activity resulting from VTA backpropagation. In ChR2-eYFP mice, we implanted an optic fiber dorsal to the VTA, and an ipsilateral guide cannula for intra-LH administration of aCSF or lidocaine (33.3 µg/µl) 3 min prior to testing mice (Fig. 2B, Supplementary Fig. 5D, E). We found that intra-LH administration of lidocaine in the absence of photostimulation did not have any effect on avoidance behavior (Supplementary Fig. 5C). In addition, intra-LH administration of lidocaine did not modify avoidance induced by VTA photostimulation (Fig. 2D), indicating that the

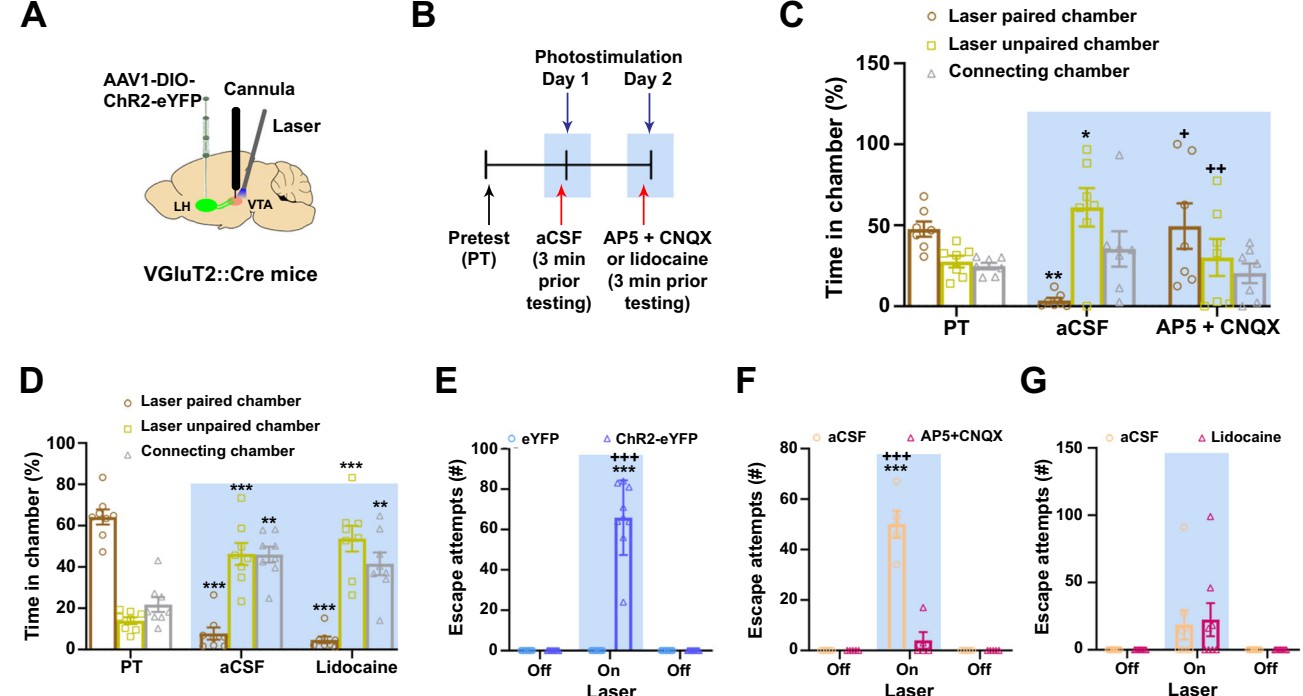

**Fig. 2 | Release of glutamate from LH-VGluT2 fibers within the VTA induces aversion and escape attempts mediated by the VTA activation of glutamate receptors. A** LH injection of AAV1-DIO-ChR2-eYFP, intra-VTA microinjection, and VTA optic fiber. **B** Timeline for microinjections experiment. **C** Intra-VTA administration of NMDA- (AP5) and AMPA- (CNQX) receptor antagonists decreased avoidance responses induced by VTA photostimulation of LH-VGluT2 fibers in ChR2-eYFP mice ($n = 7$; chamber x experimental phase: $F_{(4,24)} = 5.09$, $P = 0.004$, ANOVA with Newman-Keuls post hoc test). * $P < 0.05$, ** $P < 0.01$, against pretest; + $P < 0.05$, ++ $P < 0.01$, against aCSF. **D** Intra-LH administration of lidocaine did not affect avoidance responses induced by VTA photostimulation of LH-VGluT2 fibers in ChR2-eYFP mice ($n = 8$; chamber x experimental phase: $F_{(4,28)} = 43.47$, $P < 0.00001$, ANOVA with Newman-Keuls post hoc test). ** $P < 0.01$, *** $P < 0.001$, against pretest. **E** VTA photostimulation of LH-VGluT2 fibers induced escape attempts (measured as jumps) in ChR2-eYFP mice ($n = 9$; eYFP mice: $n = 9$; eYFP: $X^2_9 = 0.00$, $P = 1$; ChR2-eYFP: $X^2_9 = 18.00$, $P = 0.001$, Friedman ANOVA). *** $P < 0.001$, against laser off, +++ $P < 0.001$, against eYFP mice. **F** Intra-VTA administration of AP5 + CNQX decreased escape attempts (jumps) induced by VTA photostimulation of LH-VGluT2 fibers in ChR2-eYFP mice ($n = 5$; treatment x experimental phase: $F_{(2,8)} = 160.57$, $P < 0.00001$, ANOVA with Newman-Keuls post hoc test). *** $P < 0.001$, against laser off; +++ $P < 0.001$, against AP5 + CNQX. **G** Intra-VTA administration of lidocaine did not affect escape attempts (jumps) induced by VTA photostimulation of LH-VGluT2 fibers in ChR2-eYFP mice ($n = 7$; treatment x experimental phase: $F_{(2,14)} = 0.86$, $P = 0.45$, n.s, ANOVA). Light-blue rectangles indicate photostimulation. Data are presented as mean values ± SEM. Source data are provided as a Source Data file.

observed avoidance responses elicited by VTA photostimulation of LH-VGluT2 fibers are mediated by LH inputs to the VTA, rather than by LH collaterals.

To determine whether anxiety was expressed along with the avoidance elicited by VTA photostimulation of LH-VGluT2 fibers, we used a modified defensive burying test[31] in which an electrified prod was replaced with an acrylic box. Each time that the mouse touched the box, it received VTA photostimulation of LH-VGluT2 fibers (473 nm, 10-ms pulses, 20 Hz, 8 mW, Supplementary Fig. 6A). While we did not detect laser-elicited defensive burying behavior in either eYFP or ChR2-eYFP mice, we found that ChR2-eYFP mice displayed active avoidance after contact with the acrylic box. To further characterize this escape response, we delineated two identical zones, one encompassing the acrylic box (laser zone) and one in the opposite side of the testing cage (no laser zone) and analyzed mouse locomotor activity (Supplementary Fig. 6B). We found that ChR2-eYFP mice spent significantly less time in the laser zone than eYFP mice (Supplementary Fig. 6C). In addition, they traveled shorter distance, had shorter durations of the longest visit, and had shorter average duration of each visit into the laser zone than eYFP mice did (Supplementary Fig. 6D–F). Nonetheless, we did not detect differences between ChR2-eYFP and eYFP mice in the number of visits and the average speed in each zone (Supplementary Fig. 6G, H), indicating that ChR2-eYFP mice were not motor impaired. We have previously shown that VTA photostimulation of LH-VGluT2 fibers does not modify spontaneous anxiety, measured as the time that mice spent in the center versus the periphery of an open field arena[22]. Here, we confirmed and extended these findings (Supplementary Fig. 7A, B) by showing that VTA photostimulation of LH-VGluT2 fibers did not induce differences in locomotor parameters during habituation to the open field arena prior to VTA photostimulation (Supplementary Fig. 8). We found that the speed and total distance traveled (Supplementary Fig. 7C–D, 9A–D, F), as well as the number of visits to the periphery and center zones (Supplementary Fig. 9E) were significantly higher in ChR2-eYFP mice in response to VTA photostimulation. In addition, VTA photostimulation of LH-VGluT2 fibers did not modify the freezing time or the number of freezing episodes in either eYFP or ChR2-eYFP mice (Supplementary Fig. 7E, F). Furthermore, we observed that ChR2-eYFP mice, but not eYFP mice, jumped (elicited escape attempts) during the epoch in which the VTA photostimulation of LH-VGluT2 fibers was delivered (Fig. 2E) and that the number of escape attempts (jumps) was significantly decreased by intra-VTA administration of NMDA and AMPA receptor antagonists (Fig. 2F). In contrast, intra-LH administration of lidocaine did not modify the escape attempts induced by VTA photostimulation (Fig. 2G, Supplementary Fig. 5D, E). We found that the latency for the initiation of locomotion or escape attempts (jumps) induced by VTA photostimulation was significantly decreased in ChR2-eYFP mice compared with eYFP mice (Supplementary Fig. 9G, H). These findings indicate that aversion and escape responses observed after VTA photostimulation of LH-VGluT2 fibers are not related to an increase in anxiety. Moreover, the lack of both defensive burying and freezing behavior suggests that the defensive strategy prioritized after VTA photostimulation is escape (expressed as an increase in speed, total distance traveled, jumps and avoidance behavior) instead of freezing.

## VTA release of glutamate from LH-VGluT2 fibers disrupts feeding behavior

Given that VTA photostimulation of inputs from LH-GABAergic neurons elicits feeding[24,25], we next determined the effect of VTA photostimulation of LH-VGluT2 fibers at different frequencies of stimulation (2.5–20 Hz) on feeding behavior (Fig. 3A, B, Supplementary Table 1). We found that none of the VTA photostimulation frequencies tested affected the latency to eating initiation for standard chow in food sated ChR2-eYFP or eYFP mice (Supplementary

Fig. 10A–E). However, VTA photostimulation of LH-VGluT2 fibers at each of the tested frequencies resulted in a significant reduction in the amount of food eaten by sated ChR2-eYFP mice (Fig. 3B), a feeding condition in which the motivation to eat is minimal. We next determined in food restricted mice, which are highly motivated for food, whether VTA photostimulation of LH-VGluT2 fibers disrupted feeding. For these studies, we divided the test in two consecutive 3-min epochs in which food restricted mice were presented with a pre-weighed amount of standard chow food, and we measured the latency to eating initiation and the amount of food eaten after each epoch. After 4 days of training, we delivered optical stimulation during the first 3-min epoch of the test (Fig. 3C). We found that VTA photostimulation of LH-VGluT2 fibers significantly increased the latency to eating initiation (Fig. 3D) and decreased the amount of food eaten only during the laser-on epoch in ChR2-eYFP mice, but not in eYFP mice (Fig. 3E). Next, we tested eYFP and ChR2-eYFP mice for 30 min experimental sessions to determine the extent to which the length of the experimental session affected the increase in the latency to eating initiation or the decrease in the amount of food eaten and found that the longer session produced similar results than those observed with shorter (3 min) sessions (Supplementary Fig. 10I, J). The increase in latency to eating initiation and the decrease in the amount of food eaten were also observed in ChR2-eYFP mice presented with highly palatable chocolate pellets (Fig. 3D, E), indicating that feeding disruptions induced by VTA photostimulation of LH-VGluT2 fibers do not depend on the hedonic properties of the food. In addition, we tested feeding behavior in food restricted VGluT2::Cre mice that we injected with AAV1-DIO encoding the inhibitory opsin Halorhodopsin (Halo) fused with eYFP into the LH (Halo-mice, Supplementary Fig. 1A, D) with bilateral optic fibers implanted dorsal to the VTA (Supplementary Fig. 1E, G). By VTA detection of cFos cellular expression, we found that photoinhibition of glutamate release from LH-VGluT2 fibers decreased VTA cFos cellular expression (Supplementary Fig. 2L, M) without alterations in the latency to eating initiation or in the amount of food eaten during short (Fig. 3D, E) or long (Supplementary Fig. 10I, J) photoinhibition periods. When testing Halo-eYFP mice in the three-chamber apparatus, we found that VTA photoinhibition of LH-VGluT2 fibers was not reinforcing (Supplementary Fig. 4E), and it did not modify anxiety-like behaviors (Supplementary Fig. 7B, G, H), locomotion (Supplementary Fig. 7C, D) or freezing behavior (Supplementary Fig. 7E, F). Given that the LH has been implicated in sex differences in the regulation of feeding behavior[32], we measured the total amount of food eaten throughout the training and testing days in male and female mice under conditions of VTA photostimulation or photoinhibition of LH-VGluT2 fibers. We did not find sex differences in the amount of food eaten, irrespective of the absence or presence of photostimulation or photoinhibition (Supplementary Fig. 10F).

Next, we determined whether multiple intercalated periods of VTA photostimulation of LH-VGluT2 fibers at 20 Hz in food restricted mice also affected feeding behavior (10 trials; absent photostimulation in odd trials and present in even trials). We found that the latency to start eating was significantly increased (Supplementary Fig. 10G), and that the amount of food eaten was reduced in ChR2-eYFP, but not in eYFP mice (Supplementary Fig. 10H). Then, we examined the effects of VTA photostimulation of LH-VGluT2 fibers on food consumption in hungry mice that were already engaged in eating (Fig. 3F). We found that VTA photostimulation immediately stopped food consumption, and significantly increased the latency to re-start eating once the photostimulation was turned off in ChR2-eYFP, but not in eYFP mice (Fig. 3G). These findings indicate that VTA photostimulation of LH-VGluT2 fibers disrupts feeding in mice with different levels of motivation for food and that this effect is not sexually dimorphic. In addition, VTA photoinhibition of LH-VGluT2 fibers did not modify basal feeding behavior.

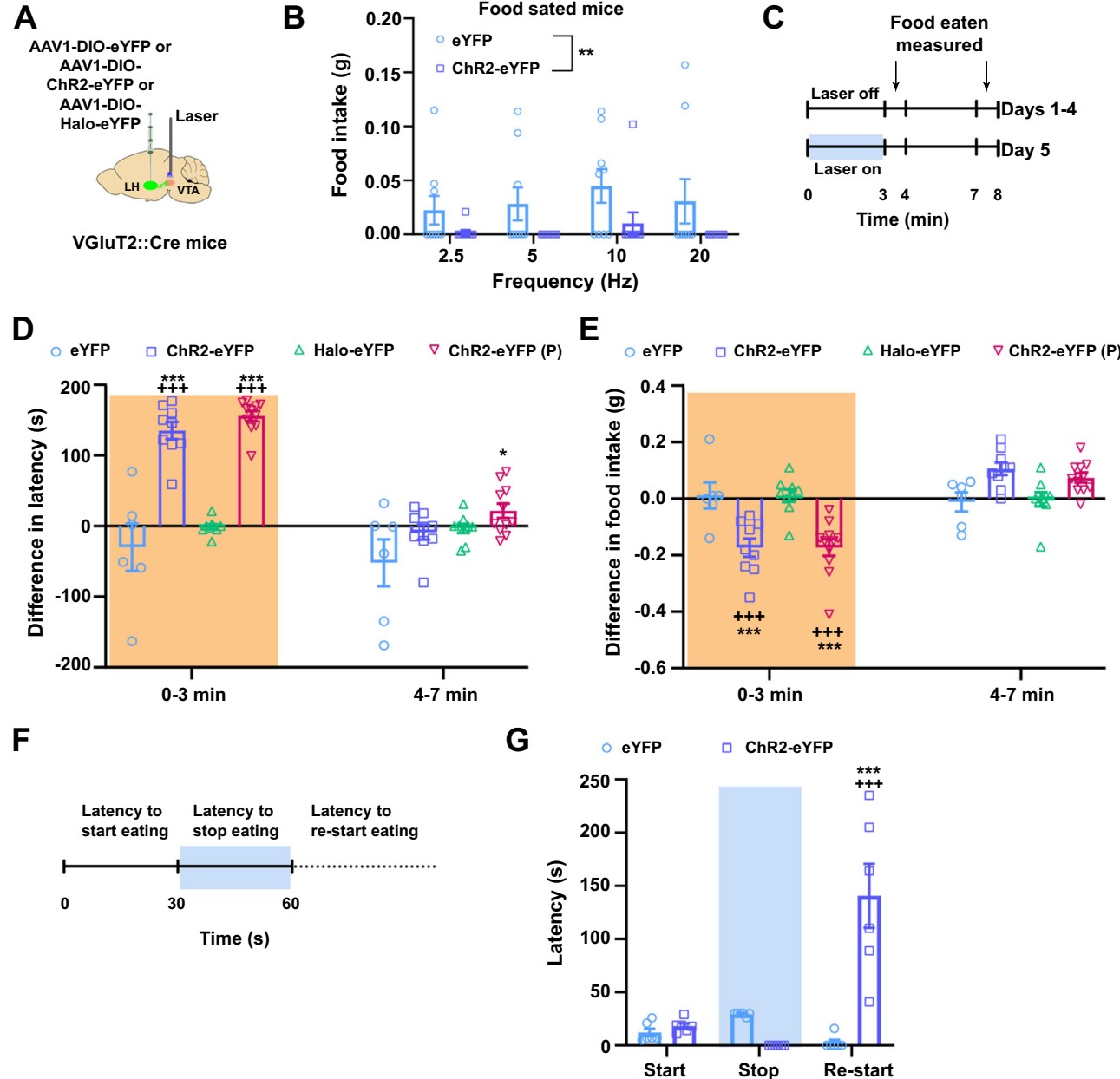

**Fig. 3 | Release of glutamate from LH-VGluT2 fibers within the VTA disrupts feeding behavior. A** LH injection of AAV1-DIO-eYFP, AAV1-DIO-ChR2-eYFP or AAV1-DIO-Halo-eYFP and VTA optic fiber. **B** Food sated ChR2-eYFP mice ($n = 10$) ate significantly less food than eYFP control mice ($n = 9$) during VTA photostimulation of LH-VGluT2 fibers regardless of stimulation frequency (group: $F_{(1,17)} = 11.06$, $P = 0.004$, ANOVA with Newman-Keuls post hoc test). **C** Timeline for feeding test in food restricted mice. **D** ChR2-eYFP mice ($n = 9$) eating standard chow or palatable pellets (ChR2-eYFP (P), $n = 11$) showed a higher latency to eating initiation during VTA laser stimulation than Halo-eYFP ($n = 9$) or eYFP control mice ($n = 6$) (group × trial: $F_{(3,31)} = 19.53$, $P < 0.00001$, ANOVA with Newman-Keuls post hoc test). * $P < 0.05$, *** $P < 0.001$, against eYFP mice, +++ $P < 0.001$, against laser off. **E** ChR2-eYFP mice ($n = 9$) presented with standard chow or palatable pellets (ChR2-

eYFP (P), $n = 11$) ate less during VTA laser stimulation than Halo-eYFP ($n = 9$) or eYFP control mice ($n = 6$; group x trial: $F_{(3,31)} = 16.21$, $P < 0.00001$, ANOVA with Newman-Keuls post hoc test). *** $P < 0.001$, against eYFP mice; +++ $P < 0.001$, against laser off. **F** Timeline to test disruption of ongoing feeding behavior in food restricted mice. **G** VTA photostimulation of LH-VGluT2 fibers for 30 s disrupted ongoing eating and significantly increased the time to resume eating in ChR2-eYFP mice ($n = 6$) but not in eYFP control mice ($n = 6$; group x experimental phase: $F_{(2,20)} = 26.44$, $P < 0.00001$, ANOVA with Newman-Keuls post hoc test). *** $P < 0.001$, against eYFP, +++ $P < 0.001$, against laser on. Light-blue rectangles indicate photostimulation, orange rectangles indicate either photostimulation or photoinhibition. Data are presented as mean values ± SEM. Source data are provided as a Source Data file.

## LH-VGluT2 neurons innervating VTA-VGluT2 neurons signal the presence of a predator and drive feeding disruption

As detailed above, we found that VTA release of glutamate from LH-VGluT2 fibers disrupted feeding behavior in hungry mice. Based on these observations, together with our previous results demonstrating that LH-VGluT2 neurons projecting to the VTA encode responses to innate threats[22], we hypothesize that the observed disruption in

feeding represents a behavioral response to innate threats inducing fear or stress, such as the presence of a predator that would induce cessation of feeding. To test this hypothesis, we first examined whether LH-VGluT2 neurons innervating the VTA signal the presence of a rat, a natural predator of mice[33]. To measure $Ca^{2+}$ activity in LH-VGluT2 neurons innervating the VTA, we drove the expression of GCaMP6s in LH-VGluT2 neurons by VTA injection of either retrograde herpes

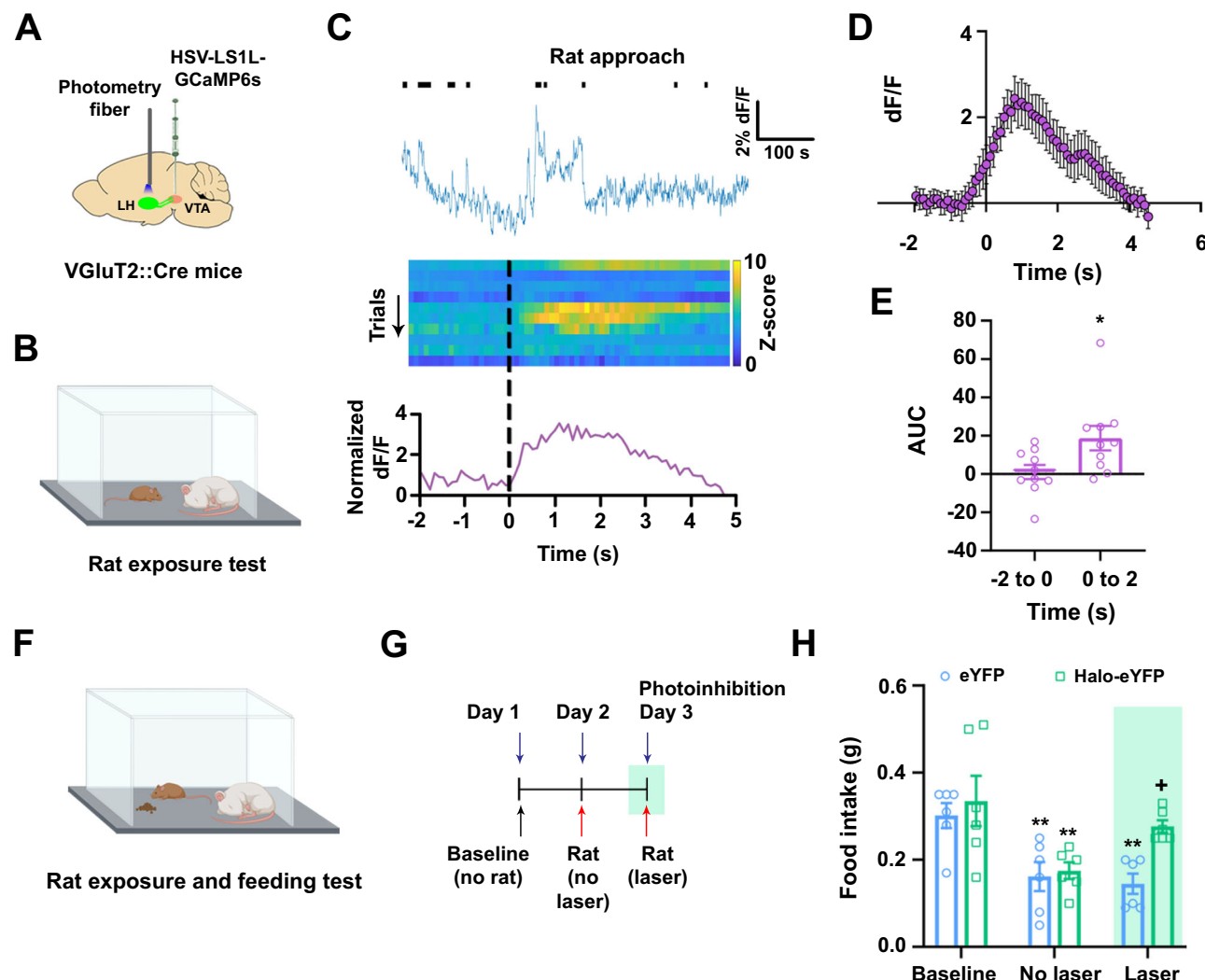

**Fig. 4 | LH-VGluT2 neurons innervating the VTA signal the presence of a predator resulting in feeding disruption. A** VTA injection of retrograde vector HSV-LS1L-GCaMP6s and LH photometry fiber. **B** Mice were tested in the presence of an anesthetized rat. **C** Whole session recording of LH-VGluT2 neurons projecting to the VTA showing approaches to the anesthetized rat (top); heatmap of $Ca^{2+}$ activity over successive rat approach trials (middle); cell population responses to rat approach onset showing increases in $Ca^{2+}$ activity in LH-VGluT2 neurons projecting to the VTA (bottom). **D** Population $Ca^{2+}$ activity (+SEM) in LH-VGluT2 neurons projecting to VTA during rat approach onset ($n = 10$). **E** AUC for $Ca^{2+}$ activity in LH-VGluT2 neurons projecting to VTA before (−2 to 0 s) and after (0 to 2 s) onset of rat

approach ($n = 10$; $t_{(18)} = -2.38$, ∗$P = 0.03$, two-tailed $t$ test). Food restricted Halo-eYFP and eYFP mice exposure to both anesthetized rat and food (**F**) and test timeline (**G**). **H** Food intake was significantly reduced in Halo-eYFP ($n = 6$) and eYFP control mice ($n = 6$) when presented with an anesthetized rat in 15-min experimental sessions. Photoinhibition restored feeding to baseline levels in Halo-eYFP mice (group x experimental phase: $F_{(2,20)} = 3.52$, $P = 0.049$, ANOVA with Newman-Keuls post hoc test). ** $P < 0.01$, against baseline, + $P < 0.05$, against eYFP. Green rectangle indicates photoinhibition. Data are presented as mean values ± SEM. Source data are provided as a Source Data file.

simplex (HSV, Fig. 4A, Supplementary Fig. 11) or retrograde AAV-GCaMP7s Cre-dependent viral vectors (Supplementary Fig. 12) in the VTA of VGluT2::Cre mice (Supplementary Fig. 12A). By electrophysiological recordings, we confirmed that LH neurons infected with HSV were healthy during calcium recordings (Supplementary Fig. 11). By in vivo fiber photometry, we measured $Ca^{2+}$ activity in LH-VGluT2 neurons projecting to the VTA in response to the presence of an anesthetized rat, a well-established method to study mouse defensive behaviors[34,35] (Fig. 4B, Supplementary Fig. 1H), and found that LH-VGluT2 neurons innervating the VTA increased their $Ca^{2+}$ activity each time that mice approached the anesthetized rat (Fig. 4C–E, Supplementary Fig. 12C-D). This increase in $Ca^{2+}$ activity was not observed when mice were presented with and approached a toy rat (Supplementary Fig. 12F–H), although mice spent more time exploring the toy rat compared to the real rat (Supplementary Fig. 12I), suggesting that the circuit is activated by specific threats and not by novelty. Given that

VTA photoinhibition of LH-VGluT2 fibers did not seem to modulate basal feeding, we next determined the extent to which LH-VGluT2 neurons projecting to the VTA play a role in fear-induced feeding disruption. For these studies, we measured feeding behavior in food restricted eYFP control and Halo-eYFP mice that had access to food in the presence of an anesthetized rat (Fig. 4F). We found that in the absence of VTA photoinhibition of LH-VGluT2 fibers (Fig. 4G), both eYFP and Halo-eYFP mice significantly decreased their food intake when presented with the anesthetized rat (Fig. 4H). In contrast, we found that in the presence of the anesthetized rat, VTA photoinhibition of LH-VGluT2 fibers increased the amount of food eaten to baseline levels in Halo-eYFP mice but not in eYFP control mice (Fig. 4H). Next, we tested eYFP and Halo-eYFP mice for 30 min experimental sessions to determine the extent to which the length of the experimental session affected fear-induced feeding disruption and found that the longer session produced similar results than those observed

with shorter (15 min) sessions (Supplementary Fig. 12J). Collectively, these results indicate that LH-VGluT2 neurons projecting to the VTA play a role in signaling specific threatening stimuli that induce disruption of feeding behavior and in modulating fear-induced feeding disruption.

Given that LH-VGluT2 neurons establish synapses preferentially with VGluT2 neurons within the VTA[22], we examined whether VTA-VGluT2 neurons are involved in fear-induced feeding disruption. By recording Ca²⁺ activity in VTA-VGluT2 neurons in response to an anesthetized rat in mice expressing GCaMP6s in VTA-VGluT2 neurons (Fig. 5A, B, Supplementary Fig. 1I), we found that VTA-VGluT2 neurons were activated when mice approached the anesthetized rat (Fig. 5C–E). To further confirm the participation of VTA-VGluT2 neurons in fear-induced feeding disruption, we expressed the apoptosis-inducing protein, caspase-3 (Casp3) in VTA-VGluT2 neurons (Fig. 5F). We confirmed the neuronal specific ablation of VTA-VGluT2 neurons by the lack of VGluT2 mRNA in the VTA of caspase mice but not in control mice (Fig. 5G–K, Supplementary Fig. 13A, B). We found that in the presence of an anesthetized rat (Fig. 5L, M), there was a decrease in food intake in control mice but not in VTA-VGluT2 ablated mice, whose food intake was similar to the baseline level in the absence of the anesthetized rat (Fig. 5N). Feeding behavior in the absence of a predator (Supplementary Fig. 13C) and locomotion (Supplementary Fig. 14D) were not affected by the ablation of VTA-VGluT2 neurons. Collectively, these findings indicate a role of VTA-VGluT2 neurons in mediating fear-induced feeding disruption.

Furthermore, we tested the possible participation of VTA-TH and GABA neurons in fear-induced feeding disruption. We expressed Casp3 in VTA-TH neurons (in TH::Cre mice, Supplementary Fig. 14A) or in VTA-GABA neurons (in VGaT::Cre mice, Supplementary Fig. 14J) and confirmed their ablation by lack of TH immunoreactivity in VTA-TH ablated mice (Supplementary Fig. 13E, F, 14B–F) and lack of VGaT mRNA in VTA-VGaT ablated mice (Supplementary Fig. 13I, J, 14K–O). In these ablated and food restricted mice, we tested feeding behavior in the presence of an anesthetized rat and food access (Supplementary Fig. 14G, H, P, Q). We found that VTA-TH ablated, VTA-VGaT ablated, as well as non-ablated control mice showed similar decreases in food intake in the presence of the anesthetized rat (Supplementary Fig. 14I, R). Ablation of VTA-TH or VTA-GABA neurons did not modify locomotion (Supplementary Fig. 13H, L) or basal feeding behavior (Supplementary Fig. 13G, K). These results indicate that, in contrast to the role of VTA-VGluT2 neurons, VTA-TH or VTA-VGaT neurons do not seem to participate in fear-induced feeding disruption.

We have previously shown that LH-VGluT2 neurons preferentially target VTA-VGluT2 neurons[22]. In here, by electron microscopy and ultrastructural analysis, we confirmed these findings in mice that expressed ChR2-mCherry in LH-VGluT2 neurons and eYFP in VTA-VGluT2 neurons (Fig. 6A). We further demonstrated that LH-VGluT2 axon terminals establish multiple synapses on a single VTA-VGluT2 soma (Fig. 6B–E), indicating a strong excitatory regulation of VTA-VGluT2 neurons by lateral hypothalamic glutamatergic neurons. Next, by combination of optogenetics and genetic ablation of VTA-VGluT2 neurons, we determined the functional connectivity between LH-VGluT2 neurons and VTA-VGluT2 neurons in mediating escape responses and fear-induced feeding disruption. For these studies, we drove the expression of ChR2-eYFP in LH-VGluT2 neurons and we drove in the same mice the expression of mCherry or caspase in VTA-VGluT2 neurons (Fig. 7A). We tested these mice in an open field arena and found that VTA photostimulation of LH-VGluT2 fibers did not modify spontaneous anxiety in mCherry- or caspase-expressing mice (Fig. 7B). While mice with intact VTA-VGluT2 neurons (mCherry mice) increased their total distance traveled and speed in the open field arena during VTA photostimulation of LH-VGluT2 fibers (Fig. 7C, D), increases of locomotion were absent in mice with genetically ablated VTA-VGluT2 neurons (caspase mice, Fig. 7C, D). Furthermore, decreases in the amount of food eaten resulting from VTA photostimulation of LH-VGluT2 fibers (either in the presence or absence of an anesthetized rat) were not observed or were attenuated in caspase mice (Fig. 7E). These findings further support a causal role of LH-VGluT2 neurons projecting to VTA-VGluT2 neurons in selecting escape responses over feeding behavior in the presence of threatening stimuli.

## Discussion

When animals perceive threats, they display complex, often non-overlapping responses such as fighting, fleeing, or freezing[36]. While there have been great advances in the identification of the neuro-circuitry mediating defensive behaviors, the neurocircuitry mediating specific aspects of defensive behaviors remains to be determined. For instance, the neuronal circuitry that mediates the switch between feeding and specific defensive behaviors is largely unexplored even though the animal's ability to make this switch is highly advantageous from an evolutionary perspective, as it preserves the organism from harm, enhancing fitness and survival. Here, we demonstrated that a subset of VTA-glutamatergic neurons selectively mediates escape (fleeing) instead of freezing by a circuitry that involves inputs from LH-glutamatergic neurons. We further demonstrated that LH-glutamatergic inputs to VTA-glutamatergic neurons play a critical role in the switch from ongoing eating to escape behavior. Thus, we demonstrated an unanticipated neuronal circuitry between LH-glutamatergic inputs to VTA-glutamatergic neurons that play a role in prioritizing escape, instead of freezing, and in the switch from feeding to escape.

### LH-glutamatergic inputs to VTA-glutamatergic neurons play a role in escape

The role of the LH in defensive behavior has been proposed for more than 50 years based on pioneer lesion, pharmacological and electrical stimulation studies[37–39]. While initial studies proposed a role of LH-excitatory projections in conveying noxious and threat information to the medulla[40], more recent studies have shown that LH-excitatory projections to specific brain structures play a role in specific aspects of defensive behavior. For instance, it has been documented that the release of LH-glutamate within the lateral habenula[28,29] and the VTA[22,27] promotes active avoidance, and the release of LH-glutamate within the periaqueductal gray promotes escape[30]. In addition to LH, the VTA has been implicated in defensive behaviors based on rat VTA lesion studies showing disrupted escape behavior[41]. A role of interactions between the VTA and LH in escape behavior was initially suggested from cat studies showing that simultaneous electrical stimulation of the LH and the VTA increases escape latency[42]. In a recent study, we demonstrated that VTA-glutamatergic neurons encode innate defensive behavior by a mechanism that involves multiple hypothalamic excitatory inputs onto single VTA-glutamatergic neurons that convey threatening information[22]. Indeed, we have shown that within the VTA, axons from LH-glutamatergic neurons establish multiple excitatory synapses on soma and dendrites of glutamatergic neurons and establish infrequent synapses on neighboring dopaminergic neurons[22]. Here, we confirmed and extended these observations by showing that VTA release of glutamate from LH-glutamatergic inputs increases jumps and avoidance behavior, but this pathway does not promote defensive burying. In addition, we found that ablation of VTA-VGluT2 neurons abolished the increase in escape responses induced by VTA photostimulation of LH-glutamatergic inputs. Thus, these findings indicate that the defensive strategy prioritized by activation of the LH-VTA glutamatergic pathway is escape instead of freezing. Furthermore, we determined that activation or inhibition of LH-glutamatergic inputs to VTA glutamate neurons does not play a role in anxiety, which has been shown to be negatively correlated with escape behavior[43,44].

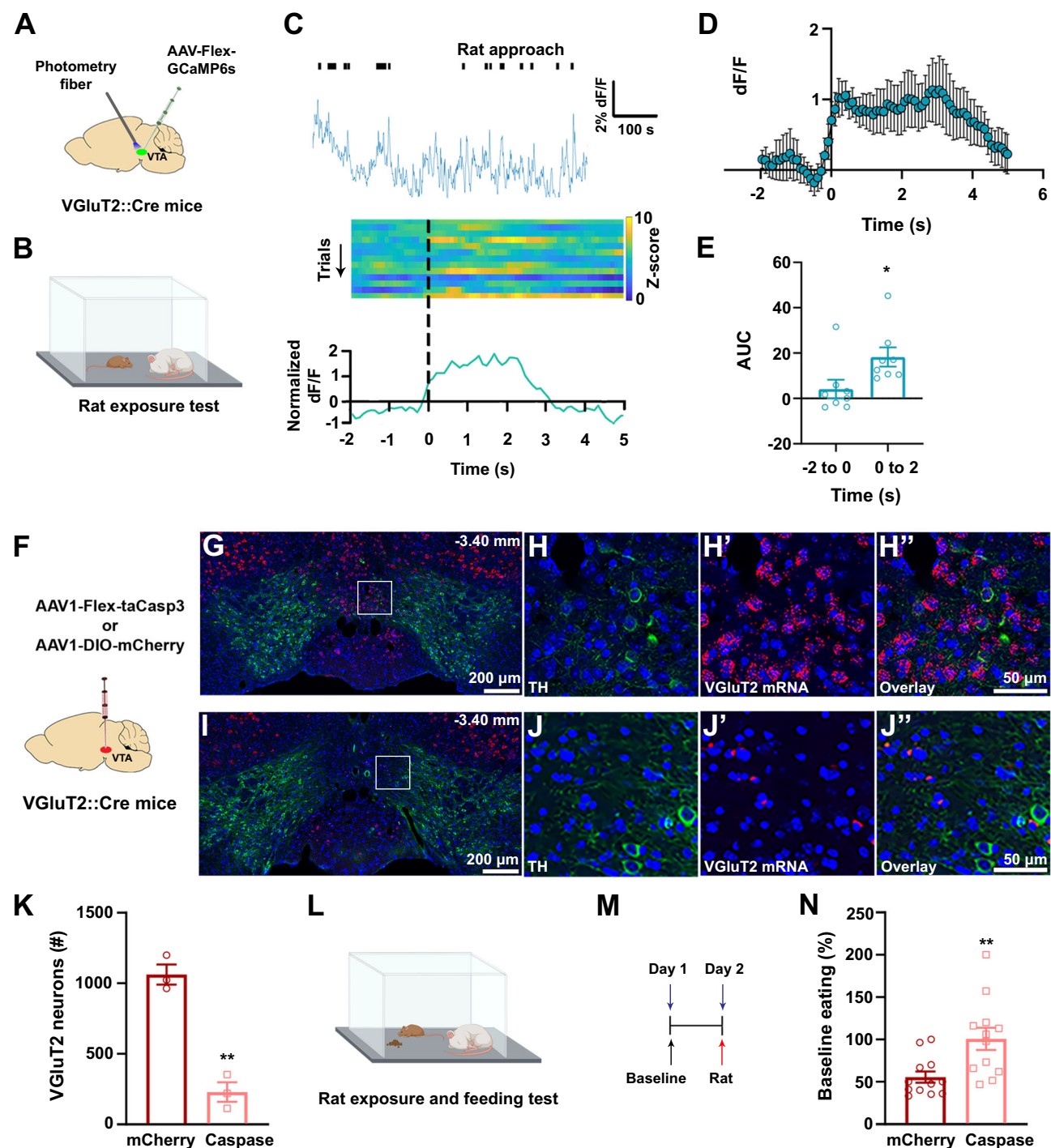

**Fig. 5 | VTA-VGluT2 neurons signal the presence of a predator resulting in feeding disruption. A** VTA injection of AAV-Flex-GCaMP6s and photometry fiber. **B** Mice were tested in the presence of an anesthetized rat. **C** Whole session recording of VTA-VGluT2 neurons showing approaches to the anesthetized rat (top); heatmap of Ca²⁺ activity over successive rat approach trials (middle); cell population responses to rat approach onset showing increases in Ca²⁺ activity in VTA-VGluT2 neurons (bottom). **D** Population Ca²⁺ activity (+SEM) in VTA-VGluT2 neurons during rat approach onset ($n = 8$). **E** AUC for Ca²⁺ activity in VTA VGluT2 neurons before (−2 to 0 s) and after (0 to 2 s) onset of the rat approach ($n = 8$; $t_{(14)} = -2.40$, *$P = 0.03$, two-tailed $t$ test). **F** VTA injection of AAV1-Flex-taCasp3 or AAV1-DIO-mCherry. Low (**G**) and high (**H**–**H″**) magnification of VTA from a control mouse (injected with AAV1-DIO-mCherry) showing neurons expressing VGluT2

mRNA (red) intermixed with TH-immunoreactive neurons (TH-IRs; green). Low (**I**) and high (**J**–**J″**) magnification of VTA from a mouse injected with AAV1-Flex-taCasp3 showing TH-IRs and lack of VGluT2 mRNA. **K** VTA-VGluT2 neurons are present in control mice ($1061.89 \pm 70.99$; 3 mice) but are infrequent in caspase mice ($228.33 \pm 69.52$; 3 mice; 7 sections per mouse; $t_{(4)} = 8.39$, two-tailed $t$ test). ** $P = 0.001$, against control mice. Food restricted caspase and control mice exposure to both anesthetized rat and food (**L**) and test timeline (**M**). **N** Food intake was significantly reduced in control mice ($n = 12$) but not in caspase mice ($n = 12$) when presented with an anesthetized rat ($t_{(22)} = -3.08$, two-tailed $t$ test). ** $P = 0.005$, against control mice. Data are presented as mean values ± SEM. Source data are provided as a Source Data file.

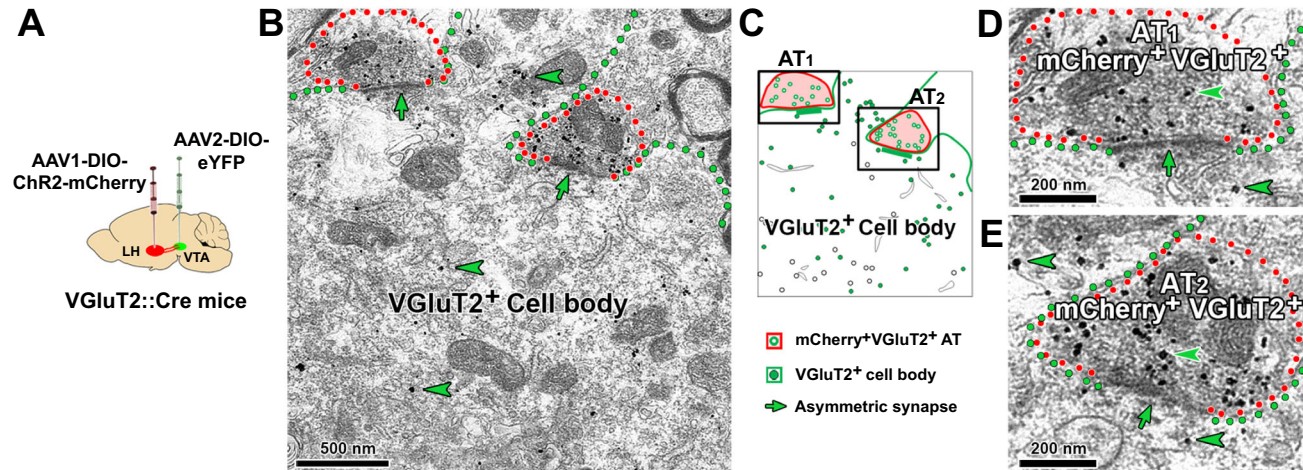

**Fig. 6 | LH-VGluT2 neurons establish synapses on VTA-VGluT2 neurons. A** LH injection of AAV2-DIO-ChR2-mCherry and VTA injection of AAV2-DIO-eYFP. **B** A VTA-VGluT2 soma (green outline with GFP detected by gold particles, arrowhead) making asymmetric synapses (green arrows) with two axon terminals (AT1–2, red outlines) from LH-VGluT2 neurons co-expressing mCherry (scattered dark material) and VGluT2 (gold particles). **C** Corresponding diagram from the image shown in (**B**). **D** AT1 at higher magnification. **E** AT2 at higher magnification.

A prior study reported a lack of optical intracranial self-stimulation (oICSS) of the LH-VTA glutamatergic pathway after one testing session[26]. While we also confirmed the lack of oICSS mediated by VTA activation of LH-glutamatergic inputs after one testing session, we found a decrease in oICSS of the LH-VTA glutamatergic pathway after extended training (10 days). In contrast to this delayed aversive response detected by the instrumental task, we found by place conditioning test that activation of the LH-VTA glutamatergic pathway resulted in immediate active avoidance and in aversion that lasted for several days. Thus, we propose that the LH-VTA glutamatergic pathway (besides being involved in innate defensive responses) plays a role in aversion with a learning component that is observed after extended training.

### LH-glutamatergic inputs to VTA play a role in the behavioral switch from ongoing eating to escape

In nature, foraging organisms must select explicit behaviors at the expense of others to fulfill their homeostatic needs, but the presence of predators often overrides feeding behavior. For instance, female mice decreased their food intake to food treated with an extract from a rat snake shed skin[45], and both laboratory and wild mice decreased their food intake to chick-peas treated with stoat or fox fecal odors[46]. In addition, while rodents avidly consume high caloric solutions (like sucrose) when given the opportunity, mice chronically exposed to the presence of rats show increases in anxiety levels with a decrease in sucrose intake[47]. This fear-induced decrease of feeding (fear-induced hypophagia) has been proposed to be mediated in part by the central amygdala[6,7]. Within this context, we have previously demonstrated that subsets of VTA-glutamatergic neurons project to the central amygdala[48], which makes it a potential downstream structure from the LH-VTA glutamatergic pathway. Regarding upstream regulation of LH neurons, a recent study[49] demonstrated that inhibition of LH by inputs from Agouti-related peptide neurons of the arcuate nucleus drove feeding in the presence of a predator (awake, hungry rat). These recent observations raised the possibility that inhibitory inputs from arcuate nucleus neurons synapsing on LH-glutamatergic neurons projecting to VTA glutamate neurons orchestrate feeding in response to innate threats.

In the present study, we found that VTA release of glutamate from LH inputs onto VTA neurons suppresses feeding in both food sated and food restricted mice and delays the re-engaging in feeding behavior. These findings indicate a role of LH-glutamatergic inputs to VTA

neurons in overriding a highly motivated behavior, such as feeding in food restricted mice. Furthermore, we demonstrated that under conditions that promote fear-induced hypophagia (mice in the presence of food and a predator), VTA photoinhibition of glutamate release from LH-glutamatergic inputs suppressed fear-induced hypophagia in food restricted mice. Collectively, these findings indicate a critical role of LH-glutamatergic inputs to VTA neurons in the switching from ongoing feeding behavior to defensive escape behavior. These findings on LH-glutamatergic inputs to VTA, together with prior findings showing that LH-glutamatergic inputs to periaqueductal gray play a role in the disruption of ongoing feeding behavior and in eliciting escape responses[30], indicate that LH-glutamatergic neurons play a crucial role as a central node that orchestrates escape responses through excitatory projections to different brain areas. Given the role of the LH-VTA glutamatergic pathway in overriding feeding in response to innate threats together with the fact that defensive behaviors in response to threats are critical for the survival of the species, it is conceivable that in addition to feeding, different ongoing behaviors (such as drinking, grooming, or mating) would be disrupted by VTA glutamate release from LH-glutamatergic fibers.

Ecological studies have shown that animals displaying beneficial action selection and trade-offs between foraging opportunities and reducing predator risk, preferentially choose safer over riskier resource use[49–52]. However, both vertebrates and invertebrates challenged with hunger will display riskier behaviors[8,53,54], such as traveling long distances away from a shelter to forage[8] or approaching predator or damaged conspecific odor cues[54,55]. In our study, we observed that mice under low food restriction levels displayed a decrease in their food intake in the presence of a predator, suggesting that in our experimental setting, mice displayed safer over riskier resource use. We demonstrated that VTA photoinhibition of glutamate release from LH-glutamatergic inputs and ablation of VTA-glutamatergic neurons (but not dopaminergic or GABAergic neurons) impaired fear-induced hypophagia. Thus, although several studies have shown that both VTA-dopaminergic and VTA-GABAergic neurons participate in different aspects of feeding behavior[56–59], we did not detect a role for VTA-dopaminergic or VTA-GABAergic neurons in fear-induced hypophagia.

In conclusion, we identified an unexpected monosynaptic brain circuit from LH-glutamatergic neurons to VTA-glutamatergic neurons that is critical in mediating the switch from ongoing feeding behavior to escape responses. While fear-induced hypophagia is adaptive and propitiates evolutionary fitness by preventing harm or death, it may

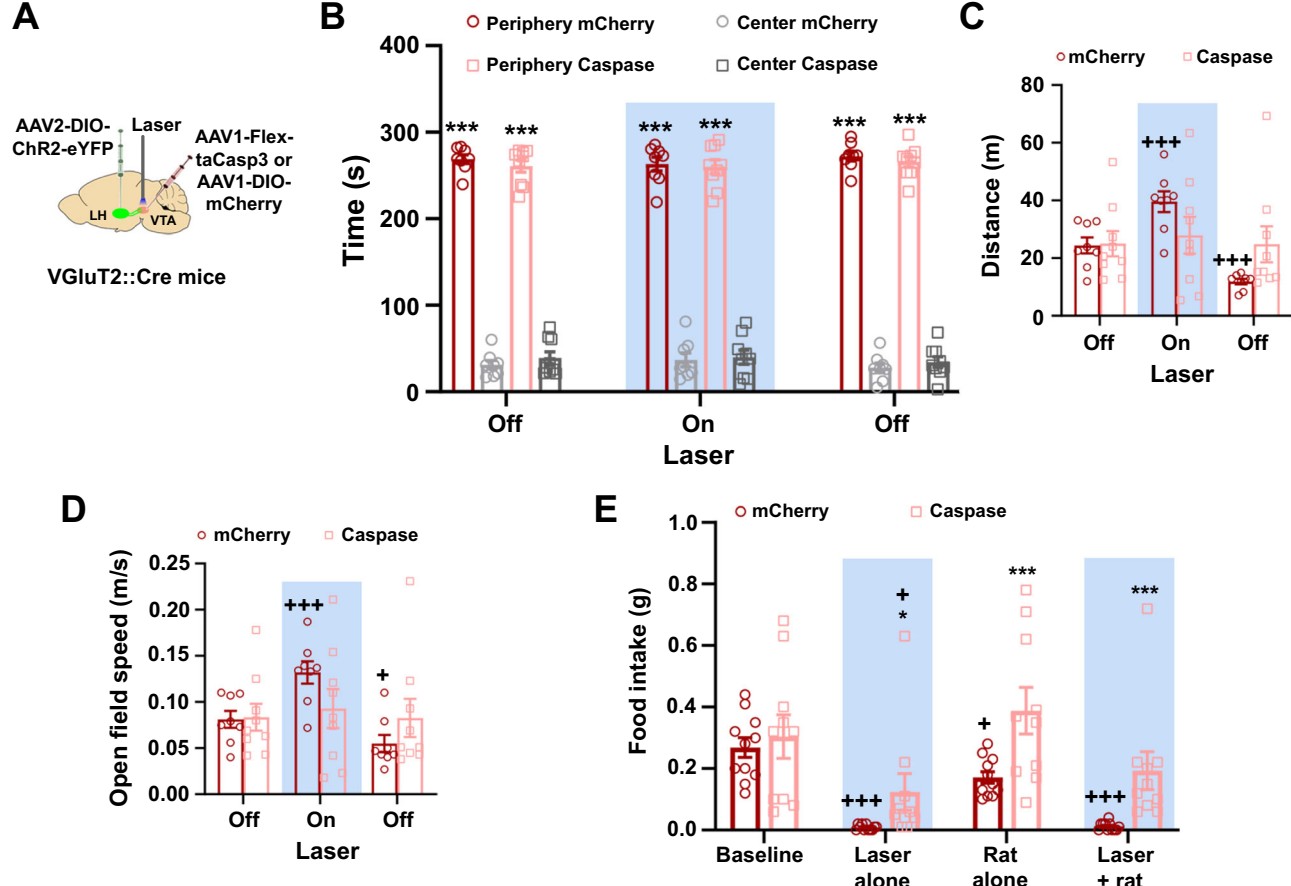

**Fig. 7 | LH-VGluT2 neurons innervating VTA-VGluT2 neurons mediate increased escape responses and predator-induced disruption of feeding. A** LH injection of AAV2-DIO-ChR2-eYFP, VTA injection of AAV1-Flex-taCasp3 or AAV1-mCherry, and VTA optic fiber. **B** Time spent in the periphery and center zones of an open field arena were similar between control mCherry ($n = 8$) and VTA-VGluT2 ablated ($n = 9$) mice before, during and after VTA photostimulation of LH-VGluT2 fibers (group x zone x experimental phase: $F_{(2,30)} = 0.07$, $P = 0.93$, n.s., ANOVA with Newman-Keuls post hoc test). *** $P < 0.001$, against center for each experimental group. **C** mCherry mice ($n = 8$), but not VTA-VGluT2 ablated mice ($n = 9$), significantly increased the distance traveled in an open field arena during VTA photostimulation of LH-VGluT2 fibers (group x experimental phase: $F_{(2,30)} = 16.47$, $P < 0.00001$, ANOVA with Newman-Keuls post hoc test). +++ $P < 0.001$, against first period of laser off. **D** mCherry mice ($n = 8$), but not VTA-VGluT2 ablated mice ($n = 9$), significantly

increased their open field speed during VTA photostimulation of LH-VGluT2 fibers (group x experimental phase: $F_{(2,30)} = 10.63$, $P = 0.0003$, ANOVA with Newman-Keuls post hoc test). + $P < 0.05$, +++ $P < 0.001$, against first period of laser off. **E** Ablation of VTA-VGluT2 neurons reversed the decrease in food intake induced by the presentation of an anaesthetized rat alone or in combination with VTA photostimulation of LH-VGluT2 fibers and partially reversed the decrease in food intake induced by VTA photostimulation of LH-VGluT2 fibers alone (mCherry, $n = 11$; Caspase, $n = 10$; group × experimental phase: $F_{(3,57)} = 3.66$, $P = 0.02$, ANOVA with Newman-Keuls post hoc test). * $P < 0.05$, *** $P < 0.001$, against mCherry mice; + $P < 0.05$, +++ $P < 0.001$, against baseline for each experimental group. Light-blue rectangles indicate photostimulation. Data are presented as mean values ± SEM. Source data are provided as a Source Data file.

lead to the onset or perpetuation of eating disorders, such as anorexia nervosa, if it becomes dysregulated and persistent. Indeed, fear of food consumption and weight gain are core symptoms and serve as diagnostic criteria in anorexia nervosa[60,61], and, when compared to healthy controls, higher levels of glutamate in the hypothalamus during fasting have been observed in some patients diagnosed with anorexia nervosa[62]. Based on these human studies and on our current results, we suggest that excessive activation of VTA-glutamatergic neurons by LH-glutamatergic projections may play a role in the development of eating disorders.

## Methods
### Animals
Male and female VGluT2::IRES::Cre mice (Slc17a6tm2(cre)Lowl/J, in C57BL/6J background from The Jackson Laboratories, Bar Harbor, ME), VGaT::Cre mice (Slc32a1tm2(cre)Lowl/J, in C57BL/6J background from The Jackson Laboratories) or TH::Cre mice (Thtm1(cre)Te/J, in C57BL/6J background from The Jackson Laboratories) were bred in the NIDA/IRP animal facility and were used in behavioral and anatomical studies.

Groups of 2–5 mice (weighing 20–30 g and at least 8 weeks old at the start of experiments) were housed in an animal vivarium maintained on a direct 12-h light-dark cycle (lights on at 7:00 am) at a constant temperature of 23 °C and 35-55% humidity. Animals were kept undisturbed for at least 1 week before the start of each experimental procedure and were handled and weighed daily to minimize handling stress during experiments. Food and water were provided ad libitum except during experimental sessions unless otherwise stated. Animal care and use were in strict accordance with institutional and international standards and were approved by the National Institute on Drug Abuse Animal Care and Use Committee (ASP 21-INRB-2). All the experiments were performed during the light phase of the diurnal cycle.

### Drugs and food
All used drugs were purchased from Sigma Aldrich (St. Louis, MO). Lidocaine, AP5, and CNQX were dissolved in artificial cerebrospinal fluid (aCSF; 124 mM NaCl, 5 mN KCl, 1.25 mM NaH2PO4, 2 mM MgSO4, 10 mM glucose, pH: 7.4). The food presented in the feeding experimental

sessions was standard laboratory chow (2018 Teklad Global 18% protein rodent diet, Harlan, Indianapolis, IN: 18.6% protein, 44.2% carbohydrate and 6.2% fat, with a caloric content of 3.1 kcal/g) or palatable chocolate pellets (F05301 Dustless precision pellets, purified, chocolate flavor, 20 mg, Bio Serv, Flemington, NJ: 18.4% protein, 59.1% carbohydrate and 5.5% fat, with a caloric content of 3.6 kcal/g).

## Surgical procedures for behavioral studies

Each mouse was anesthetized with isoflurane (2.5–3% for induction, 1.5-2% for maintenance), and placed in a stereotaxic frame where its skull was exposed and leveled. Cre-inducible adeno-associated virus (AAV, serotype 1 or 2) encoding the light-sensitive proteins channelrhodopsin-2 (ChR2) or halorhodopsin (Halo) tethered to the enhanced yellow fluorescent protein (eYFP) or eYFP alone under the control of the EF1α promoter; were used (NIDA Genetic Engineering and Viral Vector Core, Baltimore, MD or Addgene, Cambridge, MA). Briefly, 250 nl of AAV1-EF1α-DIO-hChR2(H134R)-eYFP (ChR2-eYFP mice), AAV1-EF1α-DIO-eYFP (eYFP mice), AAV1- EF1α-DIO-eNpHR3.0-eYFP (Halo-eYFP mice) were bilaterally injected into the LH of VGluT2::Cre mice at AP: −1.3, ML: ±1.0, DV: −5.2 (coordinates in mm, from bregma[63]). Injections were done with a flow rate of 100 nl/min, using an UltraMicroPump with a Micro 4 controller, 10 μl Nanofil syringes, and 35 g needles (WPI Inc., Sarasota, FL). The needle was left in place for additional 3 min to prevent reflux. For the lidocaine experiment, mice were unilaterally injected with AAV1-EF1α-DIO-hChR2(H134R)-eYFP in the LH. Additional cohorts of VGluT2::Cre, TH::Cre, or VGaT::Cre mice were injected with 200 nl of AAV1-EF1α-FLEX-taCasp3-TEVp or the control vector AAV1-EF1α-DIO-mCherry in the VTA, with medial injections for VGluT2::Cre mice and bilateral injections for TH::Cre and VGaT::Cre mice. For the calcium imaging experiments, mice were injected with 200 nl of either the Cre-dependent retrograde herpes simplex virus HSV-hEF1α-LS1L-GCaMP6s (Massachusetts General Hospital Gene Delivery Technology Core, Cambridge, MA), retro-AAV-eYFP, retro-AAV-GCaMP7s or AAV1-Syn-FLEX-GCaMP6s-WPRE-SV40 (Addgene) into the VTA. For ultrastructural analyses, 250 nl of AAV2- EF1α-DIO-ChR2-mCherry were injected unilaterally in the LH and 200 nl of AAV2-DIO-eYFP were injected in the VTA. Intracranial optical fibers or guide cannulae were implanted at least 8 weeks after viral injections. For VTA photostimulation of LH-VGluT2 fibers, mice were implanted with unilateral chronic optic fibers (200 μm diameter, BFL37-200, Thorlabs, Newton, NJ) directed just dorsal to the right VTA (AP: -3.4, ML: -0.3, DV: -4.3). For VTA photoinhibition of LH-VGluT2 fibers, mice were implanted with bilateral chronic optic fibers. For pharmacological studies, a guide cannula (22 gauge, PlasticsOne, Roanoke, VA) and an optic fiber were each lowered at a 10° angle toward the right VTA (optic fiber: AP: −3.4, ML: +0.5, DV: −4.3; cannula: AP: −3.4, ML: +1.1, DV: −4.4). For the lidocaine experiments, a guide cannula was implanted above the right LH (AP: −1.3, ML: −1.0, DV: −4.2) and an optic fiber was implanted in the right VTA (AP: −3.4, ML: −0.3, DV: −4.3). For calcium imaging studies, a 400 μm core optic fiber (photometry fiber, 0.48 NA) embedded in a 2.5 mm ferrule (Doric Lenses, Canada) was implanted either dorsal to the right VTA or dorsal to the right LH (VTA: AP: −3.2, ML: −0.3, DV: −4.1; LH: AP: −1.3, ML: −1.0, DV: −5.0). One or two stainless-steel screws and dental acrylic cement were used to anchor the optic fibers and cannula to the skull. A cap and dummy cannula were used to prevent cannula blockade. In all surgical procedures, animals were given the analgesic meloxicam (5 mg/kg) to prevent post-surgical pain or discomfort and were allowed at least 10 days of recovery before the beginning of any experimental manipulation. Body weight was measured daily after surgery to ensure proper recovery.

## Fiber photometry recordings and data analysis

For all recordings, GCaMP6s or GCaMP7s were excited at two wavelengths (490, calcium-dependent signal and 405 nm isosbestic control[64]) by amplitude modulated signals from two light-emitting diodes reflected off dichroic mirrors and coupled into the photometry fiber. Signals emitted from GCaMP6s and its isosbestic control channel then returned through the same optic fiber and were acquired using a femtowatt photoreceiver (Model 2151; Newport; Irvine, CA), digitized at 1 kHz, and then recorded by a real-time signal processor (RZ5D; Tucker Davis Technologies, TDT, Alachua, FL) running the Synapse software suite (TDT). Analysis of the resulting signal was then performed using the open source pMAT suite[65] run on MATLAB (Natick, MA) scripts. Changes in fluorescence across the experimental session (ΔF/F) were calculated by smoothing signals from the isosbestic control channel[64], scaling the isosbestic control signal by regressing it on the smoothed GCaMP signal, and then generating a predicted 405 nm signal using the linear model generated during the regression. Calcium independent signals on the predicted 405 nm channel were then subtracted from the raw GCaMP signal to remove movement, photobleaching, and fiber-bending artifacts. Signals from the GCaMP channel were then scored to normalize changes in fluorescence across animals (normalized ΔF/F). Peri-event time-locked histograms were then created by averaging changes in fluorescence (ΔF/F) across repeated trials during windows encompassing behavioral events of interest. Video recordings synchronized with neuronal acquisition clocks were acquired at 30 Hz (ANY-maze, Stoelting, Wood Dale, IL).

## Immunofluorescence and RNAscope

Mice were anesthetized with chloral hydrate (80 mg/ml) and perfused transcardially with 4% (w/v) paraformaldehyde (PFA) in 0.1 M phosphate buffer (PB), pH 7.3. Brains were left in 4% PFA for 2 h and transferred to 18% sucrose in DEPC-treated PB overnight at 4 °C. Midbrain coronal free-floating sections (VTA, 16 μm thick) were rinsed with DEPC-treated PB. Sections were incubated with mouse anti-TH antibody (1:1000, MAB318, Millipore, Burlington, MA) for 2 h at 30 °C in DEPC-treated PB supplemented with 4% BSA, 0.3% Triton X-100, and RNasin (5 μl/ml, N2115, Promega, Madison, WI, USA). After rinsing with DEPC-PB, sections were incubated with secondary donkey anti-mouse Alexa Fluor 488 or donkey anti-mouse Alexa Fluor 594 (1:100, 715-545-150 or 715-585-151, Jackson ImmunoResearch, West Grove, PA) for 1 h at 30 °C. Sections were rinsed with DEPC-treated PB, mounted onto Fisher SuperFrost slides and dried overnight at 60 °C. RNAscope in situ hybridization was performed using the RNAscope Multiplex Fluorescent v1 assay according to the manufacturer's instructions (Advanced Cell Diagnostics, Newark, CA, USA). Briefly, sections were treated with heat and protease digestion, which was followed by hybridization with a mixture containing target probes to mouse VGluT2 (319171, Advanced Cell Diagnostics, Newark, CA) or VGaT (319191, Advanced Cell Diagnostics, Newark, CA) mRNAs. Additional sections were hybridized with the bacterial gene DapB as a negative control, which did not exhibit fluorescent labeling. VGluT2 or VGaT mRNA were detected by Alexa 594. RNAscope in situ hybridization with immunolabeled sections were viewed, analyzed, and photographed with a Zeiss LSM880 confocal microscope equipped with Airyscan/CY7.5 (Zeiss, White Plains, NY). Negative control hybridizations showed negligible fluorophore expression. Neurons were counted when the stained cell was at least 5 μm in diameter. Pictures were adjusted to match contrast and brightness by using Adobe Photoshop (Adobe Systems, San Jose, CA). The number of mice (n = 3/group; 9 sections/mouse) analyzed was based upon previous studies in our lab using radioactive detection of VGluT2 mRNA from rat VTA neurons[66,67].

## cFos immunolabeling

Midbrain coronal free-floating sections (VTA, 30 μm thick) were rinsed with PB and incubated for 1 h in PB supplemented with 4% BSA and 0.3% Triton X-100. This was followed by overnight incubation at 4 °C with rabbit anti–c-Fos (1:300, Santa Cruz Biotechnology, SC-52). After being rinsed three times for 10 min each time in PB, sections were processed with an ABC kit (Vector Laboratories, Burlingame, CA). The material was incubated for 2 h at room temperature in a 1:200 dilution

of the biotinylated secondary antibody (goat anti-rabbit, BA-1000, Vector Laboratories), rinsed with PB, and incubated with avidin-biotinylated horseradish peroxidase for 1 h. Sections were rinsed, and the peroxidase reaction was then developed with 0.05% 3,3-diamino-benzidine-4 HCl and 0.03% hydrogen peroxide ($H_2O_2$). The sections were mounted on coated slides and were then photographed at 20x under bright-field illumination for cFos staining using an Olympus VS200 Scanner (Evident, Waltham, MA).

## Electron microscopy

Methods were described in a previous study[22]. Briefly, vibratome tissue sections were rinsed and incubated with 1% sodium borohydride to inactivate free aldehyde groups, rinsed and then incubated with blocking solution. Sections were then incubated with primary antibodies [mouse anti-mCherry (1:1000, Takara, #632543, Mountain View, CA), and rabbit anti-GFP (1:2000, Frontier Institute, GFP-Rb-Af2020, Hokkaido, Japan)]. All primary antibodies were diluted with 1% normal goat serum (NGS), 4% BSA in PB supplemented with 0.02% saponin and incubations were for 24 h at 4 °C. Sections were rinsed and incubated overnight at 4 °C in the corresponding secondary antibodies. Sections were rinsed in PB, and then in double-distilled water, followed by silver enhancement of the gold particles with the Nanoprobe Silver Kit (2012, Nanoprobes Inc., Yaphank, NY) for 7 min at room temperature. Next, sections were incubated in avidin-biotinylated horseradish peroxidase complex in PB for 2 h at room temperature and washed. Peroxidase activity was detected with 0.025% 3,3'-diamino-benzidine (DAB) and 0.003% $H_2O_2$ in PB for 5-10 min. Sections were rinsed with PB and fixed with 0.5% osmium tetroxide in PB for 25 min, washed in PB, followed by double-distilled water, and then contrasted in freshly prepared 1% uranyl acetate for 35 min. Sections were dehydrated through a series of graded alcohols and with propylene oxide. Afterwards, they were flat embedded in Durcupan ACM epoxy resin (14040, Electron Microscopy Sciences, Hatfield, PA). Resin-embedded sections were polymerized at 60 °C for 2 days. Sections of 60 nm were cut from the outer surface of the tissue with an ultramicrotome UC7 (Leica Microsystems, Deerfield, IL) using a diamond knife (Diatome, Hatfield, PA). The sections were collected on formvar-coated single slot grids and counterstained with Reynolds lead citrate. Sections were examined and photographed using a Tecnai G2 12 transmission electron microscope (Fei Company, Hillsboro, OR) equipped with the OneView digital micrograph camera (Gatan, Pleasanton, CA).

## Ultrastructural analysis of brain tissue

Serial ultrathin sections of the VTA (bregma −2.92 mm to −3.64 mm) were obtained from 3 VGluT2::Cre mice injected with ChR2-mCherry in the LH and with eYFP in the VTA. Synaptic contacts were classified according to their morphology and immunolabel and photographed at a magnification of 6,800-13,000×. The morphological criteria used for identification and classification of cellular components or type of synapse observed in these thin sections were as previously described[68]. Briefly, in the serial sections, a terminal containing greater than five immunogold particles was considered as immunopositive terminal. Pictures were adjusted to match contrast and brightness by using Adobe Photoshop (Adobe Systems Incorporated, Seattle, WA). Electron microscopy and confocal analysis quantification were blinded.

## Histological verification of viral expression and optic fiber placement

After termination of behavioral testing procedures, each tested mouse was deeply anesthetized with chloral hydrate (80 mg/ml) and perfused transcardially with 4% (w/v) PFA in 0.1 M PB, pH 7.3. Brains were left in 4% PFA for 2 h and transferred to 18% sucrose in DEPC-treated PB overnight at 4 °C. Cryo-coronal sections (30 μm thick) were prepared in a cryostat (CM3050 S, Leica Biosystems, Deerfield, IL) and incubated in PB supplemented with 4% BSA and 0.3% Triton X-100 for 1 h. For the

detection of eYFP, sections were incubated with the primary antibody (mouse anti-GFP, 1:500, 632381, Takara, Mountain View, CA) overnight at 4 °C. After rinsing 3 × 10 min in PB and incubation in biotinylated goat anti-mouse antibody (1:200, BA-9200, Vector Laboratories), the sections were rinsed with PB and incubated in avidin-biotinylated horseradish peroxidase (1:200, ABC kit, Vector Laboratories) for 1 h at room temperature. Sections were rinsed, and the peroxidase reaction was developed with 0.05% 3,3'-diaminobenzidine (DAB) and 0.03% $H_2O_2$. Sections were then mounted on coated slides. Brightfield images were collected with an Olympus MVX10 with 0.63× objective (Olympus, Waltham, MA).

## Electrophysiology

Eight weeks after virus injections into the VTA, mice were deeply anesthetized with isoflurane, were decapitated and their brains were quickly removed into oxygenated (95% O2/5% CO2), ice-cold high sucrose-based cutting solution (in mM): 220 sucrose, 2.5 KCl, 0.5 CaCl2, 7 MgSO4, 1.25 NaH2PO4, 26 NaHCO3, 20 glucose (pH 7.2–7.4). Coronal slices containing the LH (220 μm) were cut using a vibratome (VT1200, Leica, Nussloch, Germany), and then were transferred into an oxygenated N-Methyl-D-glucamine (NMDG)-based recovery solution at 33 °C for 7 min (in mM): 93 NMDG, 3 KCl, 10 MgSO4, 0.5 CaCl2, 1.2 NaH2PO4, 30 NaHCO3, 25 glucose, 20 HEPES, 5 sodium ascorbate, 3 sodium pyruvate (pH 7.3-7.4, ~310 mOsm-1). After recovery, slices were incubated in oxygenated artificial cerebrospinal fluid (aCSF) at room temperature (in mM): 126 NaCl, 3 KCl, 2.4 CaCl2, 1.2 NaH2PO4, 26 NaHCO3, 11 glucose, MgCl2 (pH 7.3–7.4, ~320 mOsm$^{-1}$). For electro-physiological recordings, slices were transferred to a recording chamber continuously perfused with fully oxygenated aCSF at 33 °C. Patch pipettes (4–6 MΩ) were pulled from filamented borosilicate glass capillaries (World Precision Instruments, Sarasota, FL) with a PC-100 micropipette puller (Narishige, Tokyo, Japan) and backfilled with an internal solution containing (in mM): 140 potassium gluconate, 2 NaCl, 1.5 MgCl2, 10 HEPES, 4 Mg-ATP, 0.3 Na2-GTP, 10 Tris-phospho-creatine, 0.1 ethylene glycol-bis (2-aminoethyl ether)-N,N,N',N'-tetra-acetic acid (EGTA) with 0.08-0.1 % biocytin (pH 7.2, 280-290 mOsm-1). Cells were visualized on an upright microscope using infrared differential interference contrast video microscopy. Whole-cell voltage-clamp recordings were made using a MultiClamp 700B amplifier (Molecular Devices, Sunnyvale, CA), low-pass filtered at 2 kHz and digitized at 10 kHz with pClamp 11.2 software (Molecular Devices).

## Behavioral studies

Intracranial self-stimulation, conditioned place aversion at 20 Hz, feeding in food-sated mice, defensive burying, and open field studies were carried out in the same cohort of mice. A different cohort of mice was used for pharmacology studies. A different cohort of mice was used for conditioned place aversion at 2.5 Hz, and feeding in food-restricted mice. A different cohort of mice was used for conditioned place preference, and the rat exposure and feeding test with photo-inhibition. Different, independent cohorts of mice were used for fiber photometry recordings of LH-VGluT2 neurons projecting to the VTA during the rat exposure test and after VTA ablation of VGluT2, VGaT, and TH neurons.

## Apparatus

Optical intracranial self-stimulation (oICSS) studies were conducted in sound-attenuated operant chambers (Med Associates, St. Albans City, VT) equipped with two operant response wheels, a house light, and a cue light situated above each of the two wheels. Wheel turns were monitored by MedPC software (Med Associates); each quarter-turn of the designated response wheel caused delivery of optical stimulation. Place conditioning studies were conducted in three-compartment chambers (ANY-Box, Stoelting, Wood Dale, IL) consisting of two main chambers (20 × 18 ×35 cm) with distinct wall patterns and a connecting

chamber ($20 \times 10 \times 35$ cm). Defensive burying was tested in acrylic chambers ($34 \times 25 \times 9$ cm) containing regular bedding. The open field chamber (AnyBox) was made of acrylic with opaque walls and a non-reflective base plate ($40 \times 40 \times 35$ cm). It was divided into the central area ($20 \times 20$ cm) and the periphery area. The same open field arena was used in the rat exposure test. The elevated plus maze was (Stoelting) was made of gray opaque plastic and a non-reflective base plate (lane width: 5 cm; arm length: 35 cm; wall height: 15 cm) and was elevated 50 cm from the floor. Feeding and cFos studies were conducted in the same chambers used for defensive burying experiments. For all the behavioral tests, the position of the animal was monitored via an overhead closed-circuit camera interfaced with video tracking software (AnyMaze, Stoelting). Fiber optic cables were attached via FC/PC connector to 473 nm or 532 nm lasers (OEM/Opto Engine LLC, Midvale, UT) for photostimulation or photoinhibition, respectively.

### Optical intracranial self-stimulation studies

ChR2-eYFP and eYFP mice were connected to the fiber optic cable and laser and were placed in operant chambers equipped with two response wheels (left and right) for daily 30-min self-stimulation testing. Each session began with illumination of the house light, which remained on for the entire session. Quarter-turns on one wheel ("active" wheel) activated a cue light above the wheel and caused a 2 s train of 20 Hz photostimulation (8 mW, 10 ms), followed by a 0.5 s "time-out" during which stimulation was not available. The cue light remained illuminated until the end of the time-out. Responses on the other ("inactive") wheel were without consequences. Total wheel turns in the active and inactive wheels, as well as reinforced wheel turns, were recorded. After 10 days of training at 20 Hz, ChR2-eYFP mice were tested with a range of pulse widths and pulse frequencies. On each test day, mice were given five 10-min trials with stimulation at one of four pulse widths (1.25, 2.5, 5, and 10 ms). In each of the five daily trials, the mice were tested at one of five frequencies (1.25, 2.5, 5, 10, and 20 Hz). The order of pulse widths was balanced across days; pulse frequencies were tested in ascending order for half of the mice and in descending order for the other half.

### Place conditioning studies

Place conditioning experiments were divided into three phases: pretest, conditioning, and test phases. During the 10-min pretest phase, ChR2-eYFP and eYFP mice were connected to the fiber optic cable and laser, placed in the connecting chamber, and allowed to freely explore the entire apparatus. Time spent in each compartment was measured. In the conditioning phase, the preferred chamber during the pretest was selected as the reinforced chamber: entrance to this chamber by the mouse triggered continuous trains of VTA photostimulation (10 ms, 8 mW, 2.5 or 20 Hz). The photostimulation remained on as long as the mouse was within the chamber. Entrance to the other chamber was without consequences. Each test lasted 30 min and was repeated for 4 days. During the test phase, the animals were tested 24 h after the last conditioning session for 10 min in the absence of stimulation to determine if they developed a conditioned preference/avoidance for any of the three chambers. An identical test was conducted 35 days later. Halo-eYFP mice were trained similarly, but continuous VTA photoinhibition was administered upon entrance to the preferred chamber. For pharmacological experiments, 28-gauge internal cannula injectors (PlasticsOne) were connected to Nanofil syringes through polyethylene tubing. The injection volume was 200 nl at a flow rate of 100 nl/min. The injector was left in place for additional 3 min to allow diffusion for and to prevent the possibility of reflux. ChR2-eYFP mice were tested daily in 10-min sessions for 3 days. On day 1 (pretest), mice were connected to the fiber optic cables and allowed to freely explore the apparatus. No stimulation was available during the pretest. During days 2 and 3, the experiment was like the one previously described: entrance into the preferred chamber triggered photostimulation

(10 ms, 8 mW, 20 Hz), while entrance into the non-preferred chamber was without consequences. On day 2, aCSF was infused into the VTA of ChR2-eYFP mice 3 min before testing. On day 3, a mix of AP5 (5 μg/μl) and CNQX (5 μg/μl) was infused into the VTA of ChR2-eYFP mice 3 min before testing. Another cohort of ChR2-eYFP mice was used to test the effects of AP5 and CNQX mixture alone (without photostimulation) on place conditioning. Another pharmacological study conducted as previously described was carried out in an additional cohort of ChR2-eYFP mice prepared with a cannula aimed at the right LH. In this experiment, lidocaine (33.3 μg/μl) was infused in the VTA of these mice 3 min before testing. Effects of intra-LH lidocaine in the absence of VTA photostimulation were investigated in an additional cohort of ChR2-eYFP mice.

### Modified defensive burying studies

ChR2-eYFP and eYFP mice were habituated to the testing environment for three days before any experimental manipulation. During the 15-min test session, each mouse was connected to the fiber optic cable and was placed in the acrylic chamber with clean bedding. Instead of the classical electrified shock-prod, we employed a small ($5.5 \times 7.5 \times 2$ cm) acrylic box, located in an area designated as the "laser zone". Each time the mouse touched the acrylic box, trains of continuous stimulation (10 ms, 8 mW, 20 Hz) were delivered until the mouse ceased contact with the acrylic box. Burying behavior was evaluated by measuring the height of bedding covering the acrylic box. A zone with identical dimensions to the acrylic box was calculated on the opposite side of the chamber and designated as the 'no laser zone'. Additional locomotor measures (time in zone, visits to zone, distance traveled in zone, longest visit to zone, average duration of visit to zone, and average speed in zone) were recorded for both the laser and no laser zones.

### Open field studies

ChR2-eYFP and eYFP mice were habituated to the testing environment for 2 h. Mice were connected to the fiber optic cable and laser, but no photostimulation was delivered. Total visits and time spent in the central and periphery areas were recorded, as well as total distance traveled and average speed, Next day, three 2-min tests were conducted: before, during, and after VTA photostimulation (10 ms, 8 mW, 20 Hz). The previously mentioned measures, as well as the number of escape attempts (jumps), latency to start jumping, and latency to start moving after the initiation of VTA photostimulation were recorded for each test. In additional cohorts of eYFP, ChR2-eYFP, and Halo-eYFP mice injected with a different AAV serotype in the LH, we repeated the open field experiment and run three consecutive 5-min tests: before, during, and after VTA photostimulation (10 ms, 8 mW, 20 Hz) or photoinhibition (continuous illumination). Here, we measured the distance traveled and the average speed, as well as the total time freezing and the number of freezing episodes. For experiments combining optogenetics and genetic ablation of VTA-VGluT2 neurons, the open field was conducted in three consecutive 5-min tests, as described for the previous experiments. Time spent in the periphery and center of the arena, as well as total distance traveled, and average speed were measured during each test. In another set of experiments, two pharmacological conditions were tested in two different cohorts of ChR2-eYFP mice in the open field arena in response to VTA photostimulation of LH-VGluT2 inputs. A cohort of ChR2-eYFP mice received intra-VTA injections of aCSF on day 1 or a mix of AP5 (5 μg/μl) and CNQX (5 μg/μl) on day 2, three min before scoring the number of escape attempts (jumps). Another cohort of ChR2-eYFP mice received LH injections of aCSF on day 1 or lidocaine (33.3 μg/μl) on day 2, three min before scoring the number of escape attempts (jumps).

### Elevated plus maze studies

eYFP and Halo-eYFP mice were connected to the fiber optic cable and laser and placed in the middle part of the elevated plus maze. Three

consecutive 5-min trials were conducted: before, during, and after VTA photoinhibition (continuous illumination). Time spent in the open and closed arms was measured for each trial.

## Feeding studies

All mice were habituated to the testing environment for three days before any experimental manipulation and were weighed daily. For experiments in food sated mice, eYFP or ChR2-eYFP mice were connected to the fiber optic cable and were placed in a chamber with a pre-weighed amount of food in a ceramic bowl. Each session lasted 10 min and was divided into ten 60-s trials. During odd-numbered trials, the VTA photostimulation remained off. During even-numbered trials, VTA photostimulation was given in 10 ms pulses with laser intensity at the end of the optic fiber adjusted to 8 mW. Stimulation-induced feeding was assessed at four stimulation frequencies (2.5, 5, 10, and 20 Hz) on separate days and in a balanced sequence. For experiments in food restricted mice, eYFP, ChR2-eYFP or Halo-eYFP mice (13 females, 11 males) were food-restricted at 90% of their free-feeding weight and were placed in a chamber with a pre-weighed amount of food in a ceramic bowl for 3 days. Each session lasted 6 min and was divided into two 3-min trials. Food was weighed after every trial. On day 4, half of the mice were connected to the fiber optic cable, under laser off conditions and the other half received VTA photostimulation (20 Hz) or photoinhibition (continuous) during the first 3-min trial of the session, followed by a period of 3 min with lasers off. On day 5, the laser administration order was reversed. Half of the eYFP control mice received VTA photostimulation and the other half received VTA photoinhibition. Given that the results were not significantly different, the data from all eYFP control mice were pooled together. The results were expressed as the difference between the values obtained the laser administration day minus the values obtained under laser off conditions. Latency to initiate eating was determined as the time it took the mice to elicit the first food bite. The amount of food eaten during the experimental sessions was subtracted to the total amount of food that each mouse received per day. One or two standard chow pellets weighting the calculated daily value was provided to each mouse individually. Once the mice consumed the provided food, they were re-grouped and housed together as they usually were. Additional cohorts of food restricted eYFP, ChR2-eYFP and Halo-eYFP mice were used to determine the effects of longer (30 min) VTA laser administration on the latency to eating initiation and the amount of food eaten. For these studies, laser administration (20 Hz for photostimulation or continuous illumination for photoinhibition, 5 s on/5 s off) was counterbalanced on days 4 and 5 and the difference was calculated as the values obtained during laser on conditions minus the values obtained during laser off conditions. The experiments addressing the effect of repetitive VTA photostimulation of LH-VGluT2 fibers on feeding behavior were conducted on food restricted eYFP or ChR2-eYFP mice that had access to a pre-weighed amount of food in a ceramic bowl during a 10-min session, which was divided into ten 60-s trials. VTA photostimulation remained off during odd-numbered trials, and VTA photostimulation was given during even-numbered trials (stimulation in 10 ms pulses at 20 Hz with laser intensity at the end of the optic fiber adjusted to 8 mW). The latency to eating initiation and the food intake were measured. In different cohorts of food restricted eYFP or ChR2-eYFP mice, we measured the latency to stop and re-start eating when mice were allowed to eat for 30 s before the administration of VTA photostimulation at 10 ms pulses (8 mW) during 30 s. The latency to stop and to re-engage in eating behavior was measured after the termination of VTA photostimulation.

## cFos induction

Mice were habituated to the testing environment while connected to the fiber optic cable (with the lasers off) during 10 min for 3 days. On the fourth day, VTA photostimulation (10 ms, 8 mW, 20 Hz) or photoinhibition (continuous illumination) of LH-VGluT2 fibers was administered for 5 min. Mice were perfused 90 min later and brain tissue was dissected for immunohistochemical studies.

## Rat exposure test

Mice were habituated to the open field arena already described for 10 min without being connected to the fiber optic cable and laser for three days. On the test day, an anesthetized male Long Evans rat (at least 8 weeks old, obtained from the NIDA/IRP breeding facility) was placed on a paper towel against one side of the open field in an area designated as the 'rat zone'. $Ca^{2+}$ signals were measured by in vivo photometry in the LH or the VTA of VGluT2::Cre mice, and a TTL (transistor-transistor logic) signal was sent to the Synapse software suite each time the mice entered the 'rat zone'. For the analysis of photometry data, the area under the curve (AUC) for the baseline window was collected from −2 s to 0 s relative to the rat approach onset (when the mouse entered the 'rat zone'). The same measures were collected for the onset window (from 0 s to 2 s relative to the rat approach onset) and were used for statistical analysis. For experiments in mice injected with retro-AAV-GCaMP7s, we repeated the photometry recordings as described for the previous experiments but the AUC for the baseline window was collected from −5 s to 0 s relative to the rat/toy approach onset and from 0 s to 5 s for the onset window. For these studies, a toy rat of the same size and appearance of a real rat was used on the first day and a real rat was used the next day. In addition, the time each mouse spent in the "rat/toy zone" or the zone opposite to the "rat/toy zone" ("opposite zone", located against the other side of the open field arena) was calculated. For the rat exposure test combined with feeding, Halo-eYFP and eYFP mice were food-restricted at 90% of their free-feeding weight and were habituated to the open field arena containing a pre-weighed amount of food in a ceramic bowl (located against one wall of the apparatus) for 15 min for three days. Next day, the amount of food eaten was measured while the mice were connected to the fiber optic cable and laser (off) to establish an eating baseline. The following day, an anesthetized rat was placed on a paper towel against the opposite wall from the ceramic bowl, and the amount of food eaten was measured again in the absence of VTA photoinhibition. On the third experimental day, VTA photoinhibition (continuous light) was administered throughout the test in the presence of the anesthetized rat, and the amount of food eaten was measured once again. The same experimental design, with sessions lasting 30 min, was conducted in additional cohorts of eYFP and Halo-eYFP mice (photoinhibition: continuous illumination, 5 s on/5 s off). For the rat exposure test combined with feeding in genetically ablated mice, testing was conducted after 3 days of habituation, when the baseline amount of food eaten was determined for food-restricted control and ablated mice during 15 min. The following day, the anesthetized rat was placed on a paper towel against the opposite wall from the ceramic bowl, and the amount of food eaten was measured again. For experiments in genetically VTA-VGluT2 ablated mice, where combination of rat exposure and VTA photostimulation were employed, mCherry control and caspase mice were habituated to the testing environment and baseline food intake was measured in a 15-min test. The effects of VTA photostimulation (20 Hz, 5 s on/5 s off) alone, presentation of the anesthetized rat alone or combination of VTA photostimulation and rat presentation on the amount of food eaten were investigated in sequential days, in which the amount of food eaten was measured.

## Statistical analyses and reproducibility

Results are presented as mean ± SEM. Latencies to start eating and jumping were analyzed using non-parametric methods due to a lack of variance for some set of data (not normal distribution). A Friedman ANOVA was employed to detect any inter-trial differences in each group, and the Mann-Whitney U-test was performed to detect between-groups differences. The rest of the data was analyzed using

parametric methods. Behavioral data were analyzed using a two-tailed Student's t-test in cases where two variables were compared. A multifactorial analysis of variance (MANOVA) with group (eYFP, ChR2-eYFP, Halo-eYFP) as the between-subjects factor, and trials, days, frequencies, intracerebral injections, wheels or phases of testing as within-subject factors, was used in cases where more than two variables were compared. When the same mice were tested under different conditions, a repeated measures ANOVA was used instead. For significant overall interactions, further analyses of partial interactions were carried out. Post hoc analyses were performed using the Newman-Keuls test when the initial p-value was significant. A result was considered significant if $P < 0.05$. All data were analyzed using Statistica software (Cloud Software Group Inc., Fort Lauderdale, FL). The statistical results for each experiment are depicted in Supplementary Table 2. All the experiments were successfully repeated three times to ensure reproducibility of the results.

### Reporting summary

Further information on research design is available in the Nature Portfolio Reporting Summary linked to this article.

## Data availability

The source data generated in this study have been deposited in the Zenodo repository under https://doi.org/10.5281/zenodo.10357212 (https://doi.org/10.5281/zenodo.10357211). Source data are provided with this paper.

## Code availability

No unreported custom computer code or algorithm was used to generate the results that are described in this study.

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

## Acknowledgements

This work was supported by the Intramural Research Program of the National Institute on Drug Abuse. We acknowledge the Genetic Engineering and Viral Vector Core, NIDA IRP, for manufacturing viral vectors used in this study.

## Author contributions

M.M. and M.F.B. conceptualized the project. M.F.B., E.C., O.E., and U.M. performed behavioral and pharmacological studies and data analysis. M.F.B., S.Z., and B.L. performed neuroanatomical studies. S.Z. performed anatomical and ultrastructural studies and data analysis. S.H. performed ex vivo recordings and data analysis. M.F.B., E.C., and U.M. performed photometry experiments and data analysis. O.E., U.M., and Y.A.B. quantified neuronal phenotypes. M.M. and M.F.B. prepared the manuscript with contributions from all co-authors.

## Competing interests

The authors declare no competing interests.
