## [Peer Review File · Nature Communications]

Lateral hypothalamic glutamatergic inputs to VTA
glutamatergic neurons mediate prioritization of innate
defensive behavior over feedingREVIEWER COMMENTS

Reviewer #1 (Remarks to the Author):

The manuscript from Barbano et al. reports a novel glutamatergic LH-VTA circuit that enables the switching between feeding to defensive behaviors. The findings in this manuscript are informative, exciting and novel, demonstrating a novel circuit basis for switching between appetitive and defensive behaviors. Although the manuscript is well-written and experimentally well-done, there are some important concerns regarding the methodology used to label neurons, missing statistical analysis and the interpretation of the reported effects. Please see below some major and minor concerns.

Major revisions:

1- In Fig 2D as well as in Suppl. Fig. 3D, although statistics data is not shown in the figure for eYFP groups and only in the legend for the Chr2 group, it looks evident that in the eYFP group there might be a significant difference between pretest (unpaired laser ~25% Fig 2D and Suppl. 3D) and T2 (unpaired laser ~40% Fig 2D and Suppl. 3D) and T1 in Suppl. 3D (unpaired laser ~40%). Statistical data should be provided (including p-value) here and authors should provide an explanation for this effect. This is something that should be expanded and revised in all figures from the main text and supplemental data.

2- The authors use lidocaine to inhibit backpropagation of AP from LH-Vglut2-CHR2 neurons synapsing onto VTA to other LH downstream targets such as PAG and LHb. However, no controls are shown to demonstrate that indeed PAG and LHb are not activated in laser on+lidocaine subjects or that backpropagation was indeed present. A simple c-fos or ephys experiment in PAG and LHb of laser ON+lidocaine CHR2 mice would greatly strengthen the interpretation that indeed backpropagation of AP to PAG or LHb was present and are not driving the avoidance behavior observed.

3- The authors state several times in the manuscript text and discussion that they do not observe fighting or freezing, for example in this statement: "Moreover, the lack of defensive burying suggests that the defensive strategy prioritized after VTA photostimulation is escape (expressed as an increase in speed, total distance traveled, jumps and avoidance behavior) instead of fighting or freezing.". However, no experimental data suggesting the role of LH-VTA in fighting or freezing is presented. This statement should be re-written based on the presented data only or should reference the precise literature that supports these statements.

4- In Fig 2, the authors use a 3-min trial (laser ON epoch) to measure food intake. How many mg of food do sated vs fasted mice eat during this time? Raw values for mg instead of the delta in food intake should be reported. I am assuming it is mg because a 3 min interval of eating is very restrictive.

5- HSV significantly disrupts neuronal physiology (sometimes leading to hyperexcitability) due to its toxicity. This is an important technical issue and the authors should show control data determining that neuronal physiology remain intact during calcium recordings. Is there any reason the authors did not choose a conventional and commercially available retrograde AAV-DIO-gCAMP6s?

Minor:

1- Please provide in Supplementary Fig 2D a statistical visual symbol (* or #) to facilitate visualization of statistical data analysis.

2- In Fig. 2D it is evident that both eYFP and ChR2 mice prefer the laser-paired chamber. Although this does not invalidate the data, the authors should state in the methods whether any counterbalancing measures were provided during experiments to circumvent this artifact.

3- Control data for lidocaine alone in avoidance behavior (Suppl. 4) should be displayed.

4- In the main text, the authors suggest that the modified paradigm presented in Suppl. 5A is an indicative of anxiogenic behavior when it is, in my opinion, a complementary paradigm to investigate avoidance behavior. There is a lot of controversy regarding whether burying paradigms are indeed a measure of anxiety-like behavior, avoidance, compulsive-like or novelty-induced normal (naturalistic) defensive behavior. My suggestion is to change this in the main text and reference the open field paradigm as a paradigm to assess anxiety-like behavior.

5- Authors should show control data that VTA-VGLUT2 Caspase 3 does not affect baseline eating.

6- The authors state in the Discussion that "Based on our current results, we hypothesize that excessive activation of VTA-glutamatergic neurons by LH-glutamatergic projections will promote escape responses and fear at the expense of caloric intake, leading to the development of eating disorders." Although interesting, this statement is extremely speculative based on prior data presented especially because the authors do not demonstrate any acute or long-term effects of LH-VTA activation in body weight loss. I suggest the authors remove this statement or re-write it in a less speculative manner.

7- A control experiment (Ephys or cfos) showing VTA activation after LH opto-stimulation and after LH opto-inhibition should be showed in the Suppl data.

8- Cortisol response is an important readout for stress, especially when in the presence of a predator. It would be very informative to know whether LH-VTA circuit influences cortisol levels in any level and thus, allows for the engagement of eating in the presence of an anesthetized predator.

Reviewer #2 (Remarks to the Author):

The lateral hypothalamus (LH) regulates feeding behavior and defensive responses through functionally distinct projections. Recent studies have demonstrated that glutamatergic neurons in the LH preferentially project to glutamatergic neurons in the ventral tegmental area (VTA), with infrequent projections to VTA dopaminergic neurons. Here, the authors investigated the role of LH-glutamatergic inputs to VTA-glutamatergic neurons in feeding and innate defensive behaviors in mice. Using transgenic mice in combination with optogenetics and fiber photometry, the authors showed that photostimulation of LH-VGLUT2 terminals in the VTA induced real-time place avoidance characterized by a reduction in the total time mice spent in the side of the chamber associated with the laser illumination. In addition, they observed that photostimulation of LH-VGLUT2 fibers in the VTA suppresses food intake, suggesting that increased activity in this pathway induces hypophagia. Consistently, either photoinhibition of LH-VGLUT2 fibers or caspase-3 mediated ablation of VTA-GluT2 neurons rescued the decrease in food intake induced by exposure to a predator threat (an anesthetized rat), indicating that activity in LH-GluT2 inputs to VTA as well as activity in VTA glutamatergic neurons is indispensable for predator-induced hypophagia in mice. Together, the authors characterized the functional role of LH-glutamatergic inputs to VTA in suppressing feeding and promoting defensive behaviors.

This is an original study that provides important information about the neural circuits that regulate food intake during threatening situations, which may be highly relevant for understanding motivational conflict. The manuscript is clear and the experiments were well-designed and executed. While the

study is easy to understand and will add important contribution to the field, there are some important points regarding data interpretation and limitations of the viral construct used during the experiments that needs to be addressed and clarified before publication.

Major Concerns:

1) The AAV1 serotype used for several experiments across the manuscript is known for exhibiting anterograde transsynaptic properties (see Zingg et al 2017, PMID: 27989459). If that is the case in the current study too, the injections in the LH might be labeling the soma of target cells in the VTA, which would result in optogenetic manipulation of VTA neurons in addition to LH fibers in the VTA. Similarly, using an AAV1 serotype for the Flex-Casp3 may consequently induce neuronal death in downstream targets, which would affect the outcome of the neuronal ablation. The authors should investigate the presence of somatic labeling in the VTA (e.g., including higher-resolution images for Supp. Fig. 1) and a caveat should be added in the discussion session to explain or rule out this possibility.

2) Because VGluT2 neurons in the LH and VTA were separately targeted in different experiments, it still remains unclear whether LH glutamatergic synapses onto VTA glutamatergic neurons play a causal role in feeding suppression and predator-induced hypophagia. Therefore, the authors should tune down their interpretation and explain that glutamatergic neurons in the LH and glutamatergic neurons in the VTA are involved in feeding suppression and predator-induced hypophagia, but not necessarily the projections from LH directly onto VTA glutamatergic neurons. For example, the experiments using AMPA-NMDA blockers do not rule out the possibility that glutamate being released from LH is acting on non-glutamatergic VTA neurons that express AMPA/NMDA receptors. In the same way, the lack of effect on predator-induced food suppression after the Flex-Casp3 experiments ablating dopaminergic or GABAergic neurons do not exclude the possibility that the attenuation in food intake following LH-VTA photoactivation is partially mediated by these two groups of cells (despite the reduced number of synaptic connections onto these two neuronal subpopulations).

3) In the experiments using anesthetized rats as a predator stimulus, it is unclear whether the increase in calcium transients observed in LH-VGluT2 neurons are specific for the presence of a predator or if these same neurons would also respond to any sort of novel or salient cues in the chamber. A control experiment, such as a toy rat, would provide more convincing evidence that LH-VgluT2 neurons convey information about the predator to VTA neurons. If LH-VGluT2 neurons also respond to other novel or arousing stimuli (as previously shown for orexinergic neurons in this same area), the authors need to tune down their interpretation that this pathway encodes predator-related signals and instead recognize the possibility that novelty or other salient signals may also contribute to the observed hypophagia.

4) The authors need to clarify how food restriction was controlled in mice that were group housed. Although the authors indicated in the methods section that mice were food restricted at 90% of their free-feeding weight, it is unclear whether mice had continuous access to chow in their home-cages or if food was completely removed from the home-cage and mice were only fed in the testing chamber. Were the animals exposed to any sort of daily food restriction while group-housed? This is an important point that may interfere with the behavioral results because socially dominant mice will have more access to food than subordinate mice. All these details should be included in the manuscript to facilitate future studies in the field trying to replicate the findings or at least reproduce the same experiments.

5) In Figure 1D, the authors should directly compare eYFP vs. ChR2 in the same graphic instead of displaying the eYFP and ChR2 groups separated into two graphics. The results can be better represented by using bar graphics with different colors for the two experimental groups during the two laser conditions and individual dots for each animal. Data for the connecting chamber should be omitted from the graphics (or added as an inset bar) as they do not provide much information to the

readers. In addition, the author should clarify whether 2-way ANOVA or MANOVA was conducted to compare eYFP vs. ChR2 across the experimental phases, as the figure legend only stated ANOVA.

6) In Figures 2D and 2E, the results for the eYFP control group appear to be duplicated in both graphics, which is not an appropriate practice. Instead, the authors should present the control group in one single bar, resulting in a total of 3 bar graphics per time segment: 1) eYFP, 2) ChR2-eYFP, 3) Halo-eYFP. The authors should also acknowledge the lack of laser control for Halo group in the discussion section. Also related to the same figures, it is unclear which days were used for calculating the differences in latency and how was the behavioral variability among animals during those days? Showing individual dots to represent each animal may reveal (or exclude) the existence of subgroups of mice with different behavioral phenotypes.

7) In addition to the possible role of the central nucleus of the amygdala in mediating fear-induced hypophagia described in the discussion section, the manuscript would benefit of a new paragraph positioning the current findings in the context of a series of recent studies describing the neural circuits that regulate approach-avoidance conflict during innate or learned threats. For example, the authors could speculate more about possible upstream and downstream circuits of LH and VTA that are participating in the regulation of food intake during threat presentation.

Minor concerns:

- i. The titles used for Figure 1 and 2 are confusing. Instead, replace them by something similar to: "Release of glutamate from LH-VGluT2 fibers in the VTA disrupts feeding behavior."
- ii. In the description of Figure 2B (Results section), the term "However" should be used instead of "In contrast". The use of "in contrast" implies an opposite effect, which doesn't seem to be the case here.
- iii. Graphics representing data in line plots in Figures 1G-1K, 2G, and 3H would be better represented using bar graphs showing the individual values of each animal. Alternatively, bar graphics using lines to connect each animal across experimental phases can be used.
- iv. In all figures, the graphics should explicitly indicate whether mice were food satiated or food restricted. Schematics for the satiated or food-restricted animals should be added or at least a visible title should be included on the top of the figure.
- v. In addition to food intake, the authors should consider re-analyzing the videos to extract other defensive behaviors expressed by the mice during the predator rat exposure in the same test (e.g., freezing, stretch-attend posture, escape, rat investigation time, etc), which may reveal important additional information about anti-predator behavior.
- vi. Add numbers to the pages and lines to facilitate future communication with the reviewers.

Reviewer #3 (Remarks to the Author):

In this manuscript Barbano et al., use optogenetics to demonstrate the aversive nature of LH-VGluT2 projections to the VTA. After showing that high frequency, synchronized activation of this terminal field elicits escape and avoidance, they then show that the same manipulation suppresses food intake. Finally, they show that these neurons as well as VTA-VGluT2 neurons (which they've previously demonstrated lie downstream) respond to an anesthetized rat and acute or chronic inhibition of this signaling rescues feeding behavior in the presence of the anesthetized rat. Given the wealth of literature demonstrating this pathway evokes a negative valence, it is hard to see the advancement of our understanding of this circuit, particularly at the level of VGluT2 which surely exhibits a fair amount of heterogeneity in the LH. Beyond that, there are a number of ways data is presented and experiments are performed that are flawed in convincing the reader of the robust effects. There are key controls absent in the photometry recordings as well. It is also unclear how aligned the findings that photoactivation of LH-VGluT2 projections to the VTA leads to escape behavior but their activity seems to ramp up during approach to a theoretical threat (anesthetized rat).

1. Previous studies have established the avoidance elicited by these LH-VGluT2 including aberrant behaviors such as walking backwards, jumping ect. making much of Figure 1 confirmatory. It is also likely that these behaviors are driven by the high frequency, synchronized firing as a result of ChR2 photoactivation. It would be insightful to have an idea of the endogenous neural activity of these cells to get an idea about how artificial this stimulation paradigm is.
2. It would be useful to show some of the statistical comparisons in data form, ie. Time spent in the chambers between the two groups. The text claims the avoidance in the ChR2-eYFP mice was observed in the absence of stimulation at 24 h and 35 days after the last conditioning session but I don't see this data anywhere. Compared to what? It looks to me that the ChR2-eYFP mice who avoided the laser-paired chamber show a full reversal at T1 and T2.
3. A number of the assays run in the Supplementary Data are superfluous as they just further demonstrate avoidance during artificial activation as in Figure 1.
4. Food intake needs to be assessed over longer periods of time given the high variability between animals, especially in the sated condition. I should mention that there is no indication how long food intake was measured in the frequency titration experiment (Fig 2B). Or how 20Hz stimulation was determined for use in subsequent food restricted experiments. Please graph food intake values, not differences as this can be extremely misleading. It's also strange that different controls (intermittent blue versus constant green light) have identical data points unless they are the same animals in which case there are key controls missing. How was eating determined for latency measurements (ie. a high frame rate camera)?
5. Food intake should also be measured across days in conditional trials to control for satiation instead of epochs. For example, Trial 1 food intake without photoactivation, Trial 2 with photoactivation in a crossover design to control for ordering effects.
6. How specific is the ramping of activity to rat approach specific to the rat? Is a similar signal observed when approaching food, novel object, a fake rat, another threat such as predator odor? This is an important distinction to be made.
7. What does the retro GCaMP expression look like, there are no representative images showing the efficacy of this approach.
8. One of the hardest concepts to wrap my head around is the activation of this circuit during anesthetized rat approach when artificial activation drives jumping and escape behaviors. The authors should evaluate signal during escape similar to PMID: 33861942 and 34468312.
9. Again for the photoinhibition experiments, this would be much more convincing if the baseline, no laser, laser trials were run over say a 30 minute period on different days to control for satiety.
10. Some food intake data is raw quantification while others are displayed as % change from baseline. Please consistently show raw food intake data to ensure no strange differences between groups exist
11. Toward the end of Figure 1 the authors definitively demonstrate that artificial activation of this circuit promotes escape behaviors like jumping so is it at all surprising that this manipulation attenuates food intake? Wouldn't it suppress all motivational drives like mating, thirst, territorial defense ect. It's unclear how specific this is to actual food intake. Does this manipulation alter feeding circuits specifically?
12. What kind of effects on locomotion and/or reward processing did ablation of VTA-TH and VTA-VGAT neurons have.
13. A useful experiment would be to determine if escape behaviors and/or suppression of food intake in the presence of an anesthetized rat still linger during activation of LH-VGluT2 projections to the VTA. In mice with ablation of VTA-VGluT2 neurons.

ANSWER TO THE REVIEWER'S COMMENTS

Reviewer #1 (Remarks to the Author):

The manuscript from Barbano et al. reports a novel glutamatergic LH-VTA circuit that enables the switching between feeding to defensive behaviors. The findings in this manuscript are informative, exciting and novel, demonstrating a novel circuit basis for switching between appetitive and defensive behaviors. Although the manuscript is well-written and experimentally well-done, there are some important concerns regarding the methodology used to label neurons, missing statistical analysis and the interpretation of the reported effects. Please see below some major and minor concerns.

Major revisions:

1- In Fig 2D as well as in Suppl. Fig. 3D, although statistics data is not shown in the figure for eYFP groups and only in the legend for the Chr2 group, it looks evident that in the eYFP group there might be a significant difference between pretest (unpaired laser ~25% Fig 2D and Suppl. 3D) and T2 (unpaired laser ~40% Fig 2D and Suppl. 3D) and T1 in Suppl. 3D (unpaired laser ~40%). Statistical data should be provided (including p-value) here and authors should provide an explanation for this effect. This is something that should be expanded and revised in all figures from the main text and supplemental data.

We have now included a table in the supplementary information section (Supplementary Table 2) with detailed statistical information for all the data reported in the manuscript. Regarding the specific points raised by the reviewer, the results from the ANOVA run for the eYFP group in figure 1D for the laser-paired chamber are not significant: $F(6,48)=1.61$, $p=0.17$. The interaction between chamber and experimental phase for the eYFP group in the supplementary figure 4D is also not significant ($F(10,80)=1.91$, $p=0.06$).

2- The authors use lidocaine to inhibit backpropagation of AP from LH-Vglut2-CHR2 neurons synapsing onto VTA to other LH downstream targets such as PAG and LHb. However, no controls are shown to demonstrate that indeed PAG and LHb are not activated in laser on+lidocaine subjects or that backpropagation was indeed present. A simple c-fos or ephys experiment in PAG and LHb of laser ON+lidocaine CHR2 mice would greatly strengthen the interpretation that indeed backpropagation of AP to PAG or LHb was present and are not driving the avoidance behavior observed.

By pharmacological experiments using lidocaine, we demonstrated that backpropagation is not participating in our behavioral observations. In addition, findings from the literature indicate that LH-PAG, LH-LHb and LH-VTA pathways form independent circuits with few if any collaterals (de Jong et al., 2019).

3- The authors state several times in the manuscript text and discussion that they do not observe fighting or freezing, for example in this statement: "Moreover, the lack of defensive burying suggests that the defensive strategy prioritized after VTA photostimulation is escape (expressed as an increase in speed, total distance traveled, jumps and avoidance behavior) instead of fighting or freezing." However, no experimental data suggesting the role of LH-VTA in fighting or freezing is presented. This statement should be re-written based on the presented data only or should reference the precise literature that supports these statements.

By conducting additional behavioral studies, we found that VTA photostimulation or photoinhibition of LH-VGluT2 fibers does not play a role in freezing behavior; these new data are included in Supplementary Fig. 7E-F. As suggested by the reviewer, we re-wrote the text to include this information and we have removed the reference to fighting behavior.

4- In Fig 2, the authors use a 3-min trial (laser ON epoch) to measure food intake. How many mg of food do sated vs fasted mice eat during this time? Raw values for mg instead of the delta in food intake should be reported. I am assuming it is mg because a 3 min interval of eating is very restrictive.

We have now included in the supplementary information section a table showing the amount of food eaten (in grams) for each experimental group, for all the feeding experiments in the manuscript (Supplementary Table 1). We have kept the delta of food intake in the figures to normalize food intake values.

5- HSV significantly disrupts neuronal physiology (sometimes leading to hyperexcitability) due to its toxicity. This is an important technical issue and the authors should show control data determining that neuronal physiology remain intact during calcium recordings. Is there any reason the authors did not choose a conventional and commercially available retrograde AAV-DIO-gCAMP6s?

By additional *in vivo* recording studies, we demonstrated that LH neurons infected by HSV virus remained healthy during calcium recordings; these new findings are reported in Supplementary Fig. 11. In addition, we conducted new *in vivo* calcium recordings of LH-glutamatergic neurons in mice with intra-VTA injection of a retro-AAV-GCaMP7s (Supplementary Fig. 12A-H) and found similar neuronal activity than the one observed in HSV injected mice.

Minor:

1- Please provide in Supplementary Fig 2D a statistical visual symbol (* or #) to facilitate visualization of statistical data analysis.

The statistical symbols have been added.

2- In Fig. 2D it is evident that both eYFP and Chr2 mice prefer the laser-paired chamber. Although this does not invalidate the data, the authors should state in the methods whether any counterbalancing measures were provided during experiments to circumvent this artifact.

The selection of the preferred chamber as the laser-paired chamber was intended and was described in the Materials and Methods section (page 19, line 768: "In the conditioning phase, the preferred chamber during the pretest was selected as the reinforced chamber: entrance to this chamber by the mouse triggered continuous trains of VTA photostimulation (10 ms, 8 mW, 2.5 or 20 Hz).").

3- Control data for lidocaine alone in avoidance behavior (Suppl. 4) should be displayed.

We have now included the requested control (Supplementary Fig. 5C).

4- In the main text, the authors suggest that the modified paradigm presented in Suppl. 5A is an indicative of anxiogenic behavior when it is, in my opinion, a complementary paradigm to investigate avoidance behavior. There is a lot of controversy regarding whether burying paradigms are indeed a measure of anxiety-like behavior, avoidance, compulsive-like or novelty-induced normal (naturalistic) defensive behavior. My suggestion is to change this in the main text and reference the open field paradigm as a paradigm to assess anxiety-like behavior.

As suggested by the reviewer, we conducted a new open field experiment to assess anxiety-like behaviors and found that VTA glutamate release from LH-VGluT2 fibers did not induce anxiety-like behavior (Supplementary Fig. 7A-F), which further support our initial observations. In addition, we have conducted an elevated plus maze test for eYFP and Halo-eYFP mice to further demonstrate the absence of anxiety-like behavior during photoinhibition (Supplementary Fig. 7G-H).

5- Authors should show control data that VTA-VGLUT2 Caspase 3 does not affect baseline eating.

These results are now included as Supplementary Fig. 13C, 13G and 13K.

6- The authors state in the Discussion that "Based on our current results, we hypothesize that excessive activation of VTA-glutamatergic neurons by LH-glutamatergic projections will promote escape responses and fear at the expense of caloric intake, leading to the development of eating disorders." Although interesting, this statement is extremely speculative based on prior data presented especially because the authors do not demonstrate any acute or long-term

effects of LH-VTA activation in body weight loss. I suggest the authors remove this statement or re-write it in a less speculative manner.

Following the suggestion of the reviewer, we have re-phrased this sentence to... “Based on these human studies and on our current results, we suggest that excessive activation of VTA-glutamatergic neurons by LH-glutamatergic projections may play a role in the development of eating disorders” (page 9, line 371).

7- A control experiment (Ephys or cfos) showing VTA activation after LH opto-stimulation and after LH opto-inhibition should be showed in the Suppl data.

We have now included the requested cFos experiment, corresponding results are shown in Supplementary Fig. 2L-M.

8- Cortisol response is an important readout for stress, especially when in the presence of a predator. It would be very informative to know whether LH-VTA circuit influences cortisol levels in any level and thus, allows for the engagement of eating in the presence of an anesthetized predator.

We thank the reviewer for this suggestion, which we will implement in future studies as an additional readout for stress in animals presented with threatening stimuli.

Reviewer #2 (Remarks to the Author):

The lateral hypothalamus (LH) regulates feeding behavior and defensive responses through functionally distinct projections. Recent studies have demonstrated that glutamatergic neurons in the LH preferentially project to glutamatergic neurons in the ventral tegmental area (VTA), with infrequent projections to VTA dopaminergic neurons. Here, the authors investigated the role of LH-glutamatergic inputs to VTA-glutamatergic neurons in feeding and innate defensive behaviors in mice. Using transgenic mice in combination with optogenetics and fiber photometry, the authors showed that photostimulation of LH-VGluT2 terminals in the VTA induced real-time place avoidance characterized by a reduction in the total time mice spent in the side of the chamber associated with the laser illumination. In addition, they observed that photostimulation of LH-VGluT2 fibers in the VTA suppresses food intake, suggesting that increased activity in this pathway induces hypophagia. Consistently, either photoinhibition of LH-VGluT2 fibers or caspase-3 mediated ablation of VTA-GluT2 neurons rescued the decrease in food intake induced by exposure to a predator threat (an anesthetized rat), indicating that activity in LH-Glut2 inputs to VTA as well as activity in VTA glutamatergic neurons is indispensable for predator-induced hypophagia in mice. Together, the authors characterized the functional role of LH-glutamatergic inputs to VTA in suppressing feeding and promoting defensive behaviors.

This is an original study that provides important information about the neural circuits that regulate food intake during threatening situations, which may be highly relevant for understanding motivational conflict. The manuscript is clear and the experiments were well-designed and executed. While the study is easy to understand and will add important contribution to the field, there are some important points regarding data interpretation and limitations of the viral construct used during the experiments that needs to be addressed and clarified before publication.

Major Concerns:

1) The AAV1 serotype used for several experiments across the manuscript is known for exhibiting anterograde transsynaptic properties (see Zingg et al 2017, PMID: 27989459). If that is the case in the current study too, the injections in the LH might be labeling the soma of target cells in the VTA, which would result in optogenetic manipulation of VTA neurons in addition to LH fibers in the VTA. Similarly, using an AAV1 serotype for the Flex-Casp3 may consequently induce neuronal death in downstream targets, which would affect the outcome of the neuronal ablation. The authors should investigate the presence of somatic labeling in the VTA (e.g., including higher-resolution images for Supp. Fig. 1) and a caveat should be added in the discussion session to explain or rule out this possibility.

We appreciate that the reviewer raised this point. We carefully reviewed VTA sections of each injected mice at high magnification and while we detected fibers from LH-glutamatergic neurons in the VTA, we did not observe labeled cell bodies. In addition, as suggested, we are now providing higher magnification images showing lack of somatic labeling within the VTA in spite of strong innervation from LH-VGluT2 fibers (Supplementary Fig. 2A-L).

2) Because VGluT2 neurons in the LH and VTA were separately targeted in different experiments, it still remains unclear whether LH glutamatergic synapses onto VTA glutamatergic neurons play a causal role in feeding suppression and predator-induced hypophagia. Therefore, the authors should tune down their interpretation and explain that glutamatergic neurons in the LH and glutamatergic neurons in the VTA are involved in feeding suppression and predator-induced hypophagia, but not necessarily the projections from LH directly onto VTA glutamatergic neurons. For example, the experiments using AMPA-NMDA blockers do not rule out the possibility that glutamate being released from LH is acting on non-glutamatergic VTA neurons that express AMPA/NMDA receptors. In the same way, the lack of effect on predator-induced food suppression after the Flex-Casp3 experiments ablating dopaminergic or GABAergic neurons do not exclude the possibility that the attenuation in food intake following LH-VTA photoactivation is partially mediated by these two groups of cells (despite the reduced number of synaptic connections onto these two neuronal subpopulations).

Based on the reviewer's suggestion, we performed additional studies combining optogenetics and genetic ablation of VTA-VGluT2 neurons, demonstrating that escape responses and disruption of feeding behavior are mediated by LH-glutamatergic projections innervating VTA-glutamatergic neurons (Fig. 5F-J).

3) In the experiments using anesthetized rats as a predator stimulus, it is unclear whether the increase in calcium transients observed in LH-VGluT2 neurons are specific for the presence of a predator or if these same neurons would also respond to any sort of novel or salient cues in the chamber. A control experiment, such as a toy rat, would provide more convincing evidence that LH-VgluT2 neurons convey information about the predator to VTA neurons. If LH-VGluT2 neurons also respond to other novel or arousing stimuli (as previously shown for orexinergic neurons in this same area), the authors need to tune down their interpretation that this pathway encodes predator-related signals and instead recognize the possibility that novelty or other salient signals may also contribute to the observed hypophagia.

By conducting additional experiments, we found that LH-VGluT2 neurons projecting to the VTA are not activated by the presence of a toy rat, data shown in Supplementary Fig. 12.

4) The authors need to clarify how food restriction was controlled in mice that were group housed. Although the authors indicated in the methods section that mice were food restricted at 90% of their free-feeding weight, it is unclear whether mice had continuous access to chow in their home-cages or if food was completely removed from the home-cage and mice were only fed in the testing chamber. Were the animals exposed to any sort of daily food restriction while group-housed? This is an important point that may interfere with the behavioral results because socially dominant mice will have more access to food than subordinate mice. All these details should be included in the manuscript to facilitate future studies in the field trying to replicate the findings or at least reproduce the same experiments.

As suggested, more details on the food restriction regime have been added to the Methods section, as follows... "The amount of food eaten during the experimental sessions was subtracted to the total amount of food that each mouse received per day. One or two standard chow pellets weighting the calculated daily value was provided to each mouse individually. Once the mice consumed the provided food, they were re-grouped and housed together as they usually were" (page 21, line 836).

5) In Figure 1D, the authors should directly compare eYFP vs. Chr2 in the same graphic instead of displaying the eYFP and Chr2 groups separated into two graphics. The results can be better represented by using bar graphics with different colors for the two experimental groups during the two laser conditions and individual dots for each animal. Data for the connecting chamber should be omitted from the graphics (or added as an inset bar) as they do not provide much information to the readers. In addition, the author should clarify whether 2-way ANOVA or MANOVA was conducted to compare eYFP vs. Chr2 across the experimental phases, as the figure legend only stated ANOVA.

Following the reviewer's suggestion, we modified the type of graph to include individual values. We have also included the comparison between eYFP and ChR2-eYFP mice, including the connecting chamber data. The specific type of ANOVA conducted in each case was specified in the Methods section, as follows... "A multifactorial analysis of variance (MANOVA) with group (eYFP, ChR2-eYFP, Halo-eYFP) as the between-subjects factor, and trials, days, frequencies, intracerebral injections, wheels or phases of testing as within-subject factors, was used in cases where more than two variables were compared. When the same mice were tested under different conditions, a repeated measures ANOVA was used instead" (page 22, line 887)

6) In Figures 2D and 2E, the results for the eYFP control group appear to be duplicated in both graphics, which is not an appropriate practice. Instead, the authors should present the control group in one single bar, resulting in a total of 3 bar graphics per time segment: 1) eYFP, 2) ChR2-eYFP, 3) Halo-eYFP. The authors should also acknowledge the lack of laser control for Halo group in the discussion section. Also related to the same figures, it is unclear which days were used for calculating the differences in latency and how was the behavioral variability among animals during those days? Showing individual dots to represent each animal may reveal (or exclude) the existence of subgroups of mice with different behavioral phenotypes.

We now re-draw the figure to include the 3 original groups plus a new ChR2-eYFP group fed with palatable food. The specific details on how the difference on the latency to initiate feeding and the difference on the amount eaten were obtained can be found in the Methods section, as follows.... "On day 4, half of the mice were connected to the fiber optic cable, under laser off conditions and the other half received VTA photostimulation (20 Hz) or photoinhibition (continuous) during the first 3-min trial of the session, followed by a period of 3 min with lasers off. On day 5, the laser administration order was reversed. The results were expressed as the difference between the values obtained the laser administration day minus the values obtained under laser off conditions" (page 21, line 829). In the methods section, we specified that one half of the eYFP control mice received photostimulation and the other half received photoinhibition. Given that we did not observe differences between the two treatments, the results were pooled together, and indicated as follows "Half of the eYFP control mice received VTA photostimulation and the other half received VTA photoinhibition. Given that the results were not significantly different, the data from all eYFP control mice were pooled together" (page 21, line 832).

7) In addition to the possible role of the central nucleus of the amygdala in mediating fear-induced hypophagia described in the discussion section, the manuscript would benefit of a new paragraph positioning the current findings in the context of a series of recent studies describing the neural circuits that regulate approach-avoidance conflict during innate or learned threats. For example, the authors could speculate more about possible upstream and downstream circuits of LH and VTA that are participating in the regulation of food intake during threat presentation.

This is a great suggestion and, although we are not studying specifically approach-avoidance conflict during innate or learned threats, we have a line of research in the lab focused on investigating the upstream and downstream structures participating in the regulation of food intake during threat presentation. However, we consider that these results are beyond the scope of the current work and will be published as independent findings.

Minor concerns:

i. The titles used for Figure 1 and 2 are confusing. Instead, replace them by something similar to: "Release of glutamate from LH-VGluT2 fibers in the VTA disrupts feeding behavior."

As suggested, we changed the titles of both figures.

ii. In the description of Figure 2B (Results section), the term "However" should be used instead of "In contrast". The use of "in contrast" implies an opposite effect, which doesn't seem to be the case here.

We have now corrected the term.

iii. Graphics representing data in line plots in Figures 1G-1K, 2G, and 3H would be better represented using bar graphs showing the individual values of each animal. Alternatively, bar graphics using lines to connect each animal across experimental phases can be used.

Following the reviewer's suggestion, we are now using bar graphs showing the individual values.

iv. In all figures, the graphics should explicitly indicate whether mice were food satiated or food restricted. Schematics for the satiated or food-restricted animals should be added or at least a visible title should be included on the top of the figure.

We have now included a title on top of the figures depicting results from food sated mice (Fig. 2B, Supplementary Fig. 10B-E).

v. In addition to food intake, the authors should consider re-analyzing the videos to extract other defensive behaviors expressed by the mice during the predator rat exposure in the same test (e.g., freezing, stretch-attend posture, escape, rat investigation time, etc), which may reveal important additional information about anti-predator behavior.

We are now presenting additional data obtained from previous experiments showing freezing behavior (Supplementary Fig. 7E-F), as well as rat investigation time (Supplementary Fig. 12I).

vi. Add numbers to the pages and lines to facilitate future communication with the reviewers.

As requested, we have numbered the pages and lines of the manuscript.

Reviewer #3 (Remarks to the Author):

In this manuscript Barbano et al., use optogenetics to demonstrate the aversive nature of LH-VGlut2 projections to the VTA. After showing that high frequency, synchronized activation of this terminal field elicits escape and avoidance, they then show that the same manipulation suppresses food intake. Finally, they show that these neurons as well as VTA-VGlut2 neurons (which they've previously demonstrated lie downstream) respond to an anesthetized rat and acute or chronic inhibition of this signaling rescues feeding behavior in the presence of the anesthetized rat. Given the wealth of literature demonstrating this pathway evokes a negative valence, it is hard to see the advancement of our understanding of this circuit, particularly at the level of VGlut2 which surely exhibits a fair amount of heterogeneity in the LH. Beyond that, there are a number of ways data is presented and experiments are performed that are flawed in convincing the reader of the robust effects. There are key controls absent in the photometry recordings as well. It is also unclear how aligned the findings that photoactivation of LH-VGlut2 projections to the VTA leads to escape behavior but their activity seems to ramp up during approach to a theoretical threat (anesthetized rat).

1. Previous studies have established the avoidance elicited by these LH-VGlut2 including aberrant behaviors such as walking backwards, jumping ect. making much of Figure 1 confirmatory. It is also likely that these behaviors are driven by the high frequency, synchronized firing as a result of ChR2 photoactivation. It would be insightful to have an idea of the endogenous neural activity of these cells to get an idea about how artificial this stimulation paradigm is.

By conducting additional behavior studies, we tested different stimulation frequencies, and found that a lower stimulation frequency (2.5 Hz) also elicits active avoidance, these data are shown in Supplementary Fig. 4D. In another set of optical intracranial self-stimulation experiments, we found that frequencies as low as 1.25 Hz did not sustain optical intracranial self-stimulation (Supplementary Fig. 3B), as we observed at higher frequencies (Supplementary Fig. 3B). Findings from the literature indicate that spontaneous firing rate of LH-VGluT2 neurons innervating the VTA range from 0-13 Hz (Rossi et al., 2021).

2. It would be useful to show some of the statistical comparisons in data form, ie. Time spent in the chambers between the two groups. The text claims the avoidance in the ChR2-eYFP mice was observed in the absence of stimulation at 24 h and 35 days after the last conditioning session but I don't see this data anywhere. Compared to what? It looks to me that the ChR2-eYFP mice who avoided the laser-paired chamber show a full reversal at T1 and T2.

As we indicated in reply to point 5 for reviewer 2, we have now included statistical comparisons between eYFP and ChR2-eYFP mice (Fig. 1D). Avoidance of the laser-paired chamber in the absence of stimulation was compared to the pretest values for each group, as indicated in the figure legend with the asterisk symbol. In addition, a table with all the statistical values has been included (Supplementary Table 2).

3. A number of the assays run in the Supplementary Data are superfluous as they just further demonstrate avoidance during artificial activation as in Figure 1.

We consider that the supplementary information contributes to demonstrate the participation of the circuit in escape responses and its lack of participation in other behaviors, such as defensive burying or anxiety-like behaviors.

4. Food intake needs to be assessed over longer periods of time given the high variability between animals, especially in the sated condition. I should mention that there is no indication how long food intake was measured in the frequency titration experiment (Fig 2B). Or how 20Hz stimulation was determined for use in subsequent food restricted experiments. Please graph food intake values, not differences as this can be extremely misleading. It's also strange that different controls (intermittent blue versus constant green light) have identical data points unless they are the same animals in which case there are key controls missing. How was eating determined for latency measurements (ie. a high frame rate camera)?

Following the suggestion of the reviewer, we have now included new data showing food intake during a 30-min period (Supplementary Fig. 10I-J). As we indicated in reply to point 4 raised by reviewer 1, we have now included in the supplementary information section a table showing the amount of food eaten (in grams) for each experimental group, for all the feeding experiments in the manuscript (Supplementary Table 1). In the methods section, we specified that one half of the eYFP mice received photostimulation and the other half received photoinhibition. Given that we did not observe differences between the two groups, the results were pooled together (page 21, line 832: "Half of the eYFP control mice received VTA photostimulation and the other half received VTA photoinhibition. Given that the results were not significantly different, the data from all eYFP control mice were pooled together."). We have re-draw Fig. 2D to reflect this. More details about how we determined the latency to start eating are provided in the Methods section (page 21, line 835: "Latency to initiate eating was determined as the time it took the mice to elicit the first food bite.").

5. Food intake should also be measured across days in conditional trials to control for satiation instead of epochs. For example, Trial 1 food intake without photoactivation, Trial 2 with photoactivation in a crossover design to control for ordering effects.

Food intake was indeed measured in different days with laser administered in the first trial (or epoch) and no laser in the second trial to control for satiation in a crossover design during days 4 and 5. We have now specified this information in the Methods section, as follows "...On day 4, half of the mice were connected to the fiber optic cable, under laser off conditions and the other half received VTA photostimulation (20 Hz) or photoinhibition (continuous) during the first 3-min trial of the session, followed by a period of 3 min with lasers off. On day 5, the laser administration order was reversed. The results were expressed as the difference between the values obtained the laser administration day minus the values obtained under laser off conditions" (page 21, line 829)

6. How specific is the ramping of activity to rat approach specific to the rat? Is a similar signal observed when approaching food, novel object, a fake rat, another threat such as predator odor? This is an important distinction to be made.

As we indicated in reply to point 3 raised by reviewer 2, we have included new data showing that LH-VGluT2 neurons projecting to the VTA are not activated by the presence of a toy rat (Supplementary Fig. 12), indicating that the observed neuronal activity is specific to an innate threat and not the result of novelty. In a previous study, we demonstrated activation of LH-VGluT2 neurons innervating the VTA in response to the presence of predator odors (TMT or cat urine, Barbano et al., 2020).

7. What does the retro GCaMP expression look like, there are no representative images showing the efficacy of this approach.

We have now included a representative image showing retro-GCaMP expressions in the LH (Supplementary Fig. 11B).

8. One of the hardest concepts to wrap my head around is the activation of this circuit during anesthetized rat approach when artificial activation drives jumping and escape behaviors. The authors should evaluate signal during escape similar to PMID: 33861942 and 34468312.

We have now included new findings showing the concurrent effect of photostimulation of the circuit and rat exposure on feeding behavior (Fig. 5H).

9. Again for the photoinhibition experiments, this would be much more convincing if the baseline, no laser, laser trials were run over say a 30 minute period on different days to control for satiety.

Following the suggestion of the reviewer, we have now included new data showing food intake during a 30-min period (Supplementary Fig. 12J) during different days.

10. Some food intake data is raw quantification while others are displayed as % change from baseline. Please consistently show raw food intake data to ensure no strange differences between groups exist

As indicated for point 4, we have now included in the supplementary information section a table (Supplementary Table 1) showing the amount of food eaten (in grams) for each experimental group, for all the feeding experiments in the manuscript.

11. Toward the end of Figure 1 the authors definitively demonstrate that artificial activation of this circuit promotes escape behaviors like jumping so is it at all surprising that this manipulation attenuates food intake? Wouldn't it suppress all motivational drives like mating, thirst, territorial defense ect. It's unclear how specific this is to actual food intake. Does this manipulation alter feeding circuits specifically?

We hypothesize that defensive behaviors being critical for the survival of the species will override any ongoing behavior. Although the investigation of this hypothesis escapes the scope of the present work, we will consider it in our future projects.

12. What kind of effects on locomotion and/or reward processing did ablation of VTA-TH and VTA-VGat neurons have.

We are including new findings showing that locomotion and basal feeding were not affected by ablation of VTA-VGluT2, VTA-TH or VTA-GABA neurons (Supplementary Fig. 13C, D, G, H, K, L).

13. A useful experiment would be to determine if escape behaviors and/or suppression of food intake in the presence of an anesthetized rat still linger during activation of LH-VGluT2 projections to the VTA. In mice with ablation of VTA-VGluT2 neurons.

Following the suggestion of the reviewer, we are now including new data showing how escape behavior and suppression of food intake in the presence of an anesthetized rat are modulated by VTA photostimulation of LH-VGluT2 fibers in mice with ablation of VTA-VGluT2 neurons (Fig. 5H).

REVIEWER COMMENTS

Reviewer #1 (Remarks to the Author):

The manuscript from Barbano et al. reports a novel glutamatergic LH-VTA circuit that enables the switching between feeding to defensive behaviors. The findings in this manuscript are informative, exciting and novel, demonstrating a novel circuit basis for switching between appetitive and defensive behaviors. The manuscript is well-written and experimentally well-done, and the authors have addressed all the concerns raised during the review.

Reviewer #2 (Remarks to the Author):

In this revised version of the manuscript, Barbano and coworkers have incorporated new experiments and made important modifications in the main text and figures, which clearly strengthened the quality of the manuscript. All my major concerns were fully addressed and I don't have any additional recommendation that would significantly improve the quality of the manuscript.

Reviewer #3 (Remarks to the Author):

The authors have performed an extensive set of experiments in this manuscript concluding that lateral hypothalamic glutamatergic inputs to VTA glutamatergic neurons mediate prioritization of innate defensive behavior over feeding.

There are still a number of points I think are unconvincing but I also understand this has been a lot of work and I don't want to delay possible publication.

I do want to highlight and feel like the authors should be required to discuss that activation of this circuit likely would suppress any motivational drive; they just focus on feeding. However, this manuscript could have easily been about the "prioritization of innate defensive behavior over drinking/mating/sleep" ect. It is a bit misleading that it comes across as being specific to feeding.

I'm a bit surprised by the photometry signals observed when very few LH neurons are marked with the retro-GCaMP approach. Are the authors concerned that the majority of this signal is likely local processes as shown in the image (Sup 11B)? What are all those other peaks in the photometry traces that are not related to predator approach? Seems to be even more of those. It's still difficult to discern how specific this ramping up of activity is specific to the anesthetized rat (even though a toy rat was used as a control) and not another relevant stimulus like a conspecific (restrained or anesthetized) or food.

Does photoinhibition of LH-VGluT2 neurons projecting to the VTA increase feeding in sated animals independent of rat presence? It seems this was only done in the hungry condition when food intake is already high. If it does drive intake in the absence of a threat, it substantially impacts their conclusions. Although I'm not going to go through all the literature, I'm almost positive some group has investigated this.

How confident are the authors about the specificity of their Casp3 VGluT2 ablated mice. There are a ton of glutamatergic anatomical regions in the vicinity that may be contributing to the observed phenotype including the glutamatergic hypothalamic defense network.

Still would have been important to see if photoinhibition or ablation of LH-VGluT2 neurons projecting to the VTA can rescue feeding effects in fasted (highly-motivated food-seeking conditions) animals in the presence of a rat.

A recent paper looked at this interaction between predation and feeding (PMID: 37442130) and should be cited in the Discussion

REVIEWER COMMENTS

Reviewer #1 (Remarks to the Author):

The manuscript from Barbano et al. reports a novel glutamatergic LH-VTA circuit that enables the switching between feeding to defensive behaviors. The findings in this manuscript are informative, exciting and novel, demonstrating a novel circuit basis for switching between appetitive and defensive behaviors. The manuscript is well-written and experimentally well-done, and the authors have addressed all the concerns raised during the review.

We thank the reviewer for the constructive critiques that improved our manuscript.

Reviewer #2 (Remarks to the Author):

In this revised version of the manuscript, Barbano and coworkers have incorporated new experiments and made important modifications in the main text and figures, which clearly strengthened the quality of the manuscript. All my major concerns were fully addressed and I don't have any additional recommendation that would significantly improve the quality of the manuscript.

We thank the reviewer for the constructive critiques that improved our manuscript.

Reviewer #3 (Remarks to the Author):

The authors have performed an extensive set of experiments in this manuscript concluding that lateral hypothalamic glutamatergic inputs to VTA glutamatergic neurons mediate prioritization of innate defensive behavior over feeding.

There are still a number of points I think are unconvincing but I also understand this has been a lot of work and I don't want to delay possible publication.

I do want to highlight and feel like the authors should be required to discuss that activation of this circuit likely would suppress any motivational drive; they just focus on feeding. However, this manuscript could have easily been about the "prioritization of innate defensive behavior over drinking/mating/sleep" ect. It is a bit misleading that it comes across as being specific to feeding.

As suggested by the reviewer, additional information related to the prioritization of defensive behavior over other behaviors has been added to the discussion (page 9, line 357).

I'm a bit surprised by the photometry signals observed when very few LH neurons are marked with the retro-GCaMP approach. Are the authors concerned that the majority of this signal is likely local processes as shown in the image (Sup 11B)? What are all those other peaks in the photometry traces that are not related to predator approach? Seems to be even more of those. It's still difficult to discern how specific this ramping up of activity is specific to the anesthetized rat (even though a toy rat was used as a control) and not another relevant stimulus like a conspecific (restrained or anesthetized) or food.

Due to the retrograde nature of the viral vector employed, all the processes shown in supplementary figure 11B belong to LH glutamatergic neurons that project to the VTA, which account for 40% of the total LH neurons projecting to the VTA (Barbano et al., 2020, supplementary figure 3). Given that we obtained consistent results by using the retro-HSV-GCaMP (figure 3) and the retro-AAV-GCaMP (figure 12) viral vectors, and that we have previously showed that LH-VGluT2 neurons projecting to the VTA increase their firing in response to predator odor and a looming stimulus mimicking the approach of an aerial predator (Barbano et al., 2020, figure 6), we are confident that our findings truly reflect that LH-VGluT2 neurons projecting to the VTA increase their activity in response to innate threatening stimuli.

The higher photometry peaks observed for individual traces were all related to approach behavior towards the anesthetized rat. Given that no other stimulus was presented to the mice besides the anesthetized rat, minor peaks observed could be due to the predator (rat) odor, which we have shown that also induces neuronal activation in LH glutamatergic neurons projecting to the VTA (Barbano et al., 2020).

Our findings on the time that mice spent exploring the toy rat (supplementary figure 12F, top panel, and 12H) indicate that this is a highly relevant stimulus. However, no increase in calcium signal was observed when mice approached the toy rat, suggesting that the increase in calcium signal observed in LH-VGluT2 neurons projecting to the

VTA is specific to the anesthetized rat, as shown in traces observed in figures 12C and 12F, which belong to the same mouse presented with the anesthetized rat (Fig. 12C) or the toy rat (Fig. 12F).

Does photoinhibition of LH-VGluT2 neurons projecting to the VTA increase feeding in sated animals independent of rat presence? It seems this was only done in the hungry condition when food intake is already high. If it does drive intake in the absence of a threat, it substantially impacts their conclusions. Although I'm not going to go through all the literature, I'm almost positive some group has investigated this.

Although it has been shown that photoinhibition of LH-VGluT2 cell bodies (Jennings et al., 2013) or projections to the lateral habenula (Stamatakis et al., 2016) induce feeding preferentially for palatable food in sated mice, LH glutamatergic neurons projecting to the VTA have been shown to be mainly unresponsive to sucrose consumption when mice are sated and also to respond more to an aversive tastant, such as quinine, when compared to sucrose (Rossi et al., 2021). However, we did not find increases in food intake after VTA photoinhibition of LH-VGluT2 fibers above the levels observed in control mice in any of the experiments involving feeding and photoinhibition (figures 2E, 3H, supplementary figures 10J, 12J). Considering that at least 15 different neuronal subpopulations have been identified within the LH glutamatergic neurons (Mickelsen et al., 2019), results obtained after inhibition of the total population of LH-VGluT2 neurons vs those involved in specific pathways are unlikely to be generalizable. However, based on the results from Rossi et al. and on our own results, it is conceivable that any increase in feeding resulting from VTA photoinhibition of LH-VGluT2 fibers is secondary to the inhibition of a threat-induced internal state in the animal or, alternatively, is more related to the consumption of palatable food, as suggested by Stamatakis et al., a possibility whose investigation is beyond the scope of our study.

How confident are the authors about the specificity of their Casp3 VGluT2 ablated mice. There are a ton of glutamatergic anatomical regions in the vicinity that may be contributing to the observed phenotype including the glutamatergic hypothalamic defense network.

Based in our detailed rostrocaudal midbrain analysis of VGluT2 mRNA cellular expression in Casp3-VGluT2 ablated mice, we are confident that the selective ablation of VGluT2 neurons is restricted to the VTA. In fact, the graphs for behavioral studies show results from mice with ablation confined to the VTA and correct position of optic probe. For anatomical studies, we collected 16 µm thick coronal sections of the entire midbrain (including VTA and neighboring areas) and after immunohistochemistry and RNAscope processing and analysis of the sections from the rostrocaudal midbrain, we confirmed ablation specifically in the VTA.

We are including a file showing 2 representative images from a VTA-VGluT2 Casp3-ablated mouse at different bregmas showing the specific ablation of VGluT2 neurons within the VTA. In contrast, the same sections showed VGluT2 neurons (expressing VGluT2 mRNA) in neighboring Red Nucleus (RN), Interpeduncular Nucleus (IP) and Supramammillary Nucleus (SuM), indicating area-specific ablation due to accurate intra-VTA injections of the viral vector. In addition, for this report, no Casp3 vector was ever injected in the lateral hypothalamic area; ablation was only performed in VTA-VGluT2 neurons, which are the main target of LH-VGluT2 neurons.

Still would have been important to see if photoinhibition or ablation of LH-VGluT2 neurons projecting to the VTA can rescue feeding effects in fasted (highly-motivated food-seeking conditions) animals in the presence of a rat.

We have performed both experiments in fasted mice (figures 3H, 5J and supplementary figure 12J) and we showed that VTA photoinhibition of LH-VGluT2 fibers reverses the fear-induced hypophagia observed in the presence of an anesthetized rat during short or long trials (figure 3H and supplementary figure 12J). Genetic ablation of LH-VGluT2 neurons projecting to the VTA reverses the fear-induced hypophagia observed in the presence of the rat, as well (figure 5J).

A recent paper looked at this interaction between predation and feeding (PMID: 37442130) and should be cited in the Discussion

As suggested by the reviewer, this article has been cited in the discussion section (page 9, line 343).